# Germline variation in *ADAMTSL1* is associated with prognosis following breast cancer treatment in young women

Latha Kadalayil[1,2], Sofia Khan[3], Heli Nevanlinna [3], Peter A. Fasching[4], Fergus J. Couch[5], John L. Hopper[6], Jianjun Liu[7,8], Tom Maishman[9], Lorraine Durcan[9], Sue Gerty[9], Carl Blomqvist[10], Brigitte Rack[11], Wolfgang Janni[11], Andrew Collins[1], Diana Eccles[12] & William Tapper[1]

To identify genetic variants associated with breast cancer prognosis we conduct a meta-analysis of overall survival (OS) and disease-free survival (DFS) in 6042 patients from four cohorts. In young women, breast cancer is characterized by a higher incidence of adverse pathological features, unique gene expression profiles and worse survival, which may relate to germline variation. To explore this hypothesis, we also perform survival analysis in 2315 patients aged ≤ 40 years at diagnosis. Here, we identify two SNPs associated with early-onset DFS, rs715212 ($P_{meta} = 3.54 \times 10^{-5}$) and rs10963755 ($P_{meta} = 3.91 \times 10^{-4}$) in *ADAMTSL1*. The effect of these SNPs is independent of classical prognostic factors and there is no heterogeneity between cohorts. Most importantly, the association with rs715212 is noteworthy (FPRP < 0.2) and approaches genome-wide significance in multivariable analysis ($P_{multivariable} = 5.37 \times 10^{-8}$). Expression quantitative trait analysis provides tentative evidence that rs715212 may influence *AREG* expression ($P_{eQTL} = 0.035$), although further functional studies are needed to confirm this association and determine a mechanism.

[1] Genetic Epidemiology and Bioinformatics Research Group, Human Development and Health Academic Unit, Faculty of Medicine, Duthie Building (MP 808), University of Southampton, Southampton General Hospital, Southampton SO16 6YD, UK. [2] Faculty of Natural and Environmental Sciences, University of Southampton, Highfield Campus, Southampton SO17 1BJ, UK. [3] Department of Obstetrics and Gynaecology, University of Helsinki and Helsinki University Hospital, HelsinkiP.O. BOX 700, 00029 HUS, Finland. [4] University Breast Center Franconia, Department of Gynaecology and Obstetrics, University Hospital Erlangen, Friedrich-Alexander University Erlangen-Nuremberg, Comprehensive Cancer Center Erlangen-EMN, 91054 Erlangen, Germany. [5] Department of Laboratory Medicine and Pathology, Mayo Clinic, Rochester, Minnesota 55901, USA. [6] Centre for Epidemiology and Biostatistics, Melbourne School of Population and Global Health, The University of Melbourne, Melbourne, VIC 3010, Australia. [7] Human Genetics, Genome Institute of Singapore, 60 Biopolis Street, Singapore 138672, Singapore. [8] Yong Loo Lin School of Medicine, National University of Singapore, 12 Science Drive 2, Singapore 117549, Singapore. [9] Southampton Clinical Trials Unit, University of Southampton and University Hospital Southampton NHS Foundation Trust, Southampton SO16 6YD, UK. [10] Department of Oncology, Helsinki University Central Hospital, P.O. Box 180, FIN-00029 Helsinki, Finland. [11] Department of Gynecology and Obstetrics, University Ulm, Prittwitzstrasse 43, 89075 Ulm, Germany. [12] Cancer Sciences Division, Faculty of Medicine, University of Southampton, Southampton University Hospitals NHS Trust, Southampton SO16 6YD, UK. Andrew Collins, Diana Eccles and William Tapper contributed equally to this work. Correspondence and requests for materials should be addressed to W.T. (email: W.J.Tapper@soton.ac.uk)

Breast cancer is the second leading cause of cancer-related death in women with nearly 450,000 deaths per year worldwide, despite advances in effective chemotherapy[1]. Modern chemotherapy regimens include anthracyclines and increasingly taxanes, before and/or after surgery (http://www.cancerresearchuk.org/). Patients are stratified according to clinical and pathological characteristics of the cancer to predict prognosis, select treatment regimen and to determine appropriate surgical options for individual patients. Frequently, these decisions are made on the basis of prognostic tools and guidelines which consider tumour characteristics (size, grade and hormone receptors), nodal involvement, onset age, family history and the mutation status of high-risk genes such as BRCA1 and BRCA2[2]. These prognostic tools help to balance the benefits of therapy against their side effects. However, patients with the same tumour characteristics and treatment frequently have different outcomes which suggests that additional factors such as inherited variation may account for these differences.

Many studies have demonstrated that germline variants contribute to the aetiology of breast cancer. They include several genome-wide association studies (GWASs) which have identified nearly 100 common low-penetrance breast cancer-associated alleles (odds ratios: 1.05–1.57)[3]. These low-penetrance alleles account for ~14% of the familial risk of disease while high-penetrance mutations in genes such as BRCA1 and BRCA2 and moderate-penetrance alleles in genes such as PALB2, ATM and CHEK2 account for a further ~16% of familial risk[4].

Familial studies were among the first to indicate that inherited variants also influence breast cancer prognosis[5] and many germline variants associated with survival have been identified. For example, germline mutations in CHEK2[6] and PALB2[7] have been implicated in poor prognosis and single-nucleotide polymorphisms (SNPs) have been associated with the risk of developing either oestrogen receptor (ER)-positive[8] or -negative[9] breast cancer subtypes, which have differing outcomes. More recently, many studies including GWASs have identified SNPs

**Table 1 Clinical characteristics of patient cohorts**

| | ABCFS | HEBCS | POSH stage 1 | SUCCESS-A | POSH stage 2 | P-value |
|---|---|---|---|---|---|---|
| SNPs passing QC | | | | | | |
| Observed | 508,505 | 501,882 | 503,568 | 566,645 | 116 | |
| Imputed | 5,150,529 | 5,395,529 | 5,196,034 | 5,006,474 | 0 | |
| No. of cases passing QC | 202 | 798 | 556 | 3183 | 1303 | |
| Age at diagnosis median (range) | — | 53 (22–87) | 36 (18–40) | 54 (19–85) | 37 (20–40) | $< 2.2 \times 10^{-16}$ |
| No. of cases aged ≤ 40 at diagnosis | 202 | 119 (15%) | 556 (100%) | 337 (11%) | 1303 (100%) | |
| Deceased (all cause) | 67 (33%) | 317 (40%) | 268 (48%) | 171 (5%) | 278 (21%) | |
| Deceased and aged ≤ 40[a] | 67 (33%) | 45 (38%) | 268 (48%) | 13 (4%) | 278 (21%) | |
| OS median (IQR), years | 15.8 (14.2) | 8.0 (5.2) | 4.8 (4.5) | 4.8 (2.4) | 6.8 (3.2) | $< 2.2 \times 10^{-16}$ |
| Disease progression | — | 368 (46%) | 288 (52%) | 335 (11%) | 325 (25%) | |
| Progressed and aged ≤ 40[a] | — | 65 (55%) | 288 (52%) | 38 (11%) | 325 (25%) | |
| DFS median (IQR), years | — | 5.0 (2.7) | 3.1 (5.5) | 4.8 (2.7) | 6.3 (3.7) | $< 2.2 \times 10^{-16}$ |
| Oestrogen receptor (ER) | | | | | | |
| Positive | 91 (45%) | 515 (65%) | 193 (35%) | 2189 (69%) | 1014 (78%) | |
| Negative | 75 (37%) | 225 (28%) | 362 (65%) | 975 (31%) | 283 (22%) | $1.00 \times 10^{-13}$ |
| Missing | 36 (18%) | 58 (7%) | 1 (<1%) | 19 (1%) | 6 (< 1%) | |
| Progesterone receptor (PR) | | | | | | |
| Positive | — | — | 112 (20%) | 2018 (63%) | 713 (55%) | |
| Negative | — | — | 369 (66%) | 1143 (36%) | 328 (25%) | $1.00 \times 10^{-13}$ |
| Missing | 202 (100%) | 798 (100%) | 75 (14%) | 22 (1%) | 262 (20%) | |
| HER2 | | | | | | |
| Positive | — | 86 (11%) | 105 (19%) | 952 (30%) | 347 (27%) | |
| Negative | — | 400 (50%) | 407 (73%) | 2166 (68%) | 776 (59%) | $1.49 \times 10^{-11}$ |
| Missing | 202 (100%) | 312 (39%) | 44 (8%) | 65 (2%) | 180 (14%) | |
| Triple negative (ER, PR, HER2) | — | — | 283 (51%) | 518 (16%) | 104 (8%) | $1.00 \times 10^{-13}$ |
| Grade | | | | | | |
| 1 | — | 140 (18%) | 13 (2%) | 144 (5%) | 97 (7%) | |
| 2 | — | 306 (38%) | 90 (16%) | 1523 (48%) | 508 (39%) | |
| 3 | — | 275 (34%) | 435 (78%) | 1494 (47%) | 670 (51%) | $1.00 \times 10^{-13}$ |
| Missing | 202 (100%) | 77 (10%) | 18 (3%) | 22 (1%) | 28 (2%) | |
| Tumour size | | | | | | |
| Size (mm) average (range) | — | 25.2 (1–100) | 29.7 (0–160) | 25.1 (1–220) | 25.7 (0.5–170) | 0.038 |
| T stage | | | | | | |
| 1 | — | 381 (48%) | 227 (41%) | 1302 (41%) | 654 (50%) | |
| 2 | — | 297 (37%) | 228 (41%) | 1645 (52%) | 505 (39%) | |
| 3 | — | 50 (6%) | 45 (8%) | 172 (5%) | 80 (6%) | |
| 4 | — | 47 (6%) | 4 (1%) | 43 (1%) | 1 (< 1%) | $1.00 \times 10^{-13}$ |
| Missing | 202 (100%) | 23 (3%) | 52 (9%) | 21 (1%) | 63 (5%) | |
| Nodal metastasis | | | | | | |
| Positive | — | 441 (55%) | 271 (49%) | 2044 (64%) | 679 (52%) | |
| Negative | — | 326 (41%) | 258 (46%) | 1115 (35%) | 613 (47%) | $1.00 \times 10^{-13}$ |
| Missing | 202 (100%) | 31 (4%) | 27 (5%) | 24 (1%) | 11 (1%) | |

[a]Percentages using number of cases aged ≤ 40 years at diagnosis as the denominator
P-value for comparison between all cohorts with data (n = 3 to 5), Pearson's $X^2$ tests were used for categorical variables and Kruskal–Wallis rank sum tests were used to compare continuous traits

directly associated with breast cancer survival that are largely independent of traditional tumour prognostic factors[10]. Most of these loci have small effect sizes (hazard ratio (HR) < 1.5) and it is important to note that many loci reported by GWASs do not reach genome-wide significance which suggests that some of the previous GWASs were underpowered due to small sample size. Indeed, it has been suggested that extremely large sample sizes are needed to establish genome-wide levels of significance in breast cancer survival studies[10]. Alternatively, focussing on a smaller cohort of patients with a particular breast cancer subtype may increase power by reducing genetic heterogeneity. These genetic determinants of prognosis are important because they could improve prognostic models, aid selection of appropriate treatments, and suggest targets for new therapies. For example, tumour gene expression profiles perform equally well or better than clinicopathologic models, possibly because they reflect a larger component of germline determinants of gene expression in an established tumour than models based on clinicopathologic features[11].

The principal aim of this study is to identify genetic determinants of breast cancer prognosis using a meta-analysis of four GWASs and a fifth replication cohort. We also investigate the role of common germline variation in a subset of patients with early onset (aged ≤ 40 years at diagnosis). Although breast cancer is uncommon in young women, with only 7% of patients aged ≤ 40 years at diagnosis and 1% < 30 years[12], it represents the most frequent form of non-skin cancer in young women, accounting for ~40% of cases[13]. Furthermore, women diagnosed between the ages of 15 and 39 years have a lower 5-year survival rate (83.5%) than women aged 40 to 49 years (89.1%) (www.cancerresearchuk.org). The worse survival of early-onset cases has been attributed to a higher incidence of adverse pathological features (higher histological grade[14], more frequent ER- and progesterone receptor (PR)-negative tumours)[15,16] but multivariable analysis has demonstrated that age is an independent risk factor after adjusting for stage, treatment and tumour characteristics[12]. Analyses of gene expression profiles have shown that tumours arising in young women can be distinguished by 367 biologically relevant gene sets[16] and that ER-positive tumours in premenopausal women overexpress AREG, TFPI2, AMPH, DBX2, RP5–1054A22.3 and KLK5, and underexpress ESR1, CYP4Z1, RANBP3L, FOXD2 and PEX3[17,18]. The impact of epidemiological risk factors, such as obesity, also differs between premenopausal women, where it reduces the risk of ER-positive tumours, and postmenopausal women where it increases overall susceptibility[19,20]. These observations have led to the suggestion that, from an aetiological perspective, early-onset breast cancer may represent a different type of disease with a unique underlying biology and response to epidemiological risk factors that could be influenced by germline variation.

We identify three association signals with suggestive significance levels and without heterogeneity between cohorts (1: rs715212 and rs10963755; 2: rs12302097; and 3: rs410155). The most significant of these is rs715212, situated in intron 19 of ADAMTSL1, which increases the risk of disease progression but only in patients with early onset (aged ≤ 40 years at diagnosis). When adjusting for the known prognostic factors, this association approaches a genome-wide level of significance and is assessed as noteworthy by the false positive report probability (FPRP < 0.2). We also demonstrate that rs715212 is nominally associated with the expression of AREG. We therefore conclude that the association between ADAMTSL1 and breast cancer prognosis may involve an interaction with AREG expression, although further functional studies are needed to confirm this association and to determine the mechanism.

## Results

**Comparison of clinicopathologic features between cohorts.** Stage-1 breast cancer samples came from four cohorts from Australia (Australian Breast Cancer Family Study (ABCFS)), Helsinki (Helsinki breast cancer study (HEBCS)), the United Kingdom (Prospective Study of Outcomes in Sporadic vs. Hereditary breast cancer (POSH)) and Germany (SUCCESS-A). A further 1303 independent patients from the POSH cohort were used for replication analysis at stage-2 (Table 1, see Methods for a full description of these cohorts). The baseline characteristics among the stage-1 and validation cohorts were significantly different (Table 1). These differences are largely due to the POSH cohort, which only recruited patients with early onset (age ≤ 40 years at diagnosis), and the selection of patients with triple-negative breast cancer (TNBC) or survival extremes from the POSH cohort at stage-1. As a result, the stage-1 POSH cohort had the highest frequency of ER, PR and HER2 negativity, grade 3 tumours, larger average tumour size and the shortest median time to disease progression and mortality. Despite these differences, the ABCFS, HEBCS and POSH stage-1 cohorts were similar in terms of the incidence of disease progression and mortality. In comparison, SUCCESS-A and POSH stage-2 had lower incidences of progression and mortality. To address these differences, survival analyses at stages 1 and 2 were adjusted for ER status and, for replicating SNPs, multivariable models were constructed using pooled data from stages 1 and 2.

**Stage-1 survival analysis and meta-analysis.** Following standard quality control (QC, see Methods), 4739 patients from four cohorts ABCFS ($N = 202$), HEBCS ($N = 798$), POSH ($N = 556$) and SUCCESS-A ($N = 3183$) and 5,848,861 SNPs (813,964 observed and 5,034,897 imputed) were used for genome-wide analysis of overall survival (OS) at stage-1 (Table 1 and Supplementary Table 1). The ABCFS cohort was excluded from the early-onset and disease-free survival (DFS) analyses because data on progression were unavailable. Consequently, 4537 patients were used for DFS in all cases and 1102 patients for early-onset analyses at stage-1. Among the survival analyses performed at stage-1 there were 823 events for OS, 991 events for DFS (all cases), 326 events for early-onset OS and 391 events for early-onset DFS. According to these sample sizes and event rates, we estimated that the combined stage-1 analysis of OS and DFS had 80% power to detect common SNPs (minor allele frequency (MAF)=0.3) with modest effects (HR=1.4, $P_\alpha = 5 \times 10^{-8}$). Due to its smaller sample size, the analysis of OS and DFS in early-onset cases was estimated to have 80% power to detect SNPs with slightly larger effect sizes (OS: HR=1.7, DFS: HR=1.6, Supplementary Fig. 2).

For each cohort, the quantile–quantile (QQ) plots and low genomic inflation factors ($\lambda \leq 1.05$) for OS and DFS in all cases and in the early-onset subset demonstrate good agreement between observed and expected $P$-values until the tail of the distributions where SNPs with $P$-values $<10^{-4}$ deviated from the null distribution (Supplementary Figs. 3–5). Systematic biases such as population stratification are therefore unlikely to contribute to the significance of these SNPs. Comparison of the QQ plots for OS and DFS showed that analyses of DFS tended to identify more SNPs with low $P$-values ($P_{\text{Cox regression}} \leq 10^{-4}$) which is consistent with the larger number of events.

Following Cox regression, we used a fixed effects meta-analysis to combine evidence across the stage-1 cohorts and visualized these results in a Manhattan plot by selecting the most significant $P$-value for each SNP from the analysis of OS or DFS, in all cases and the early-onset subset (Fig. 1). Despite meta-analysis of upto 4739 patients and sufficient power to detect common SNPs

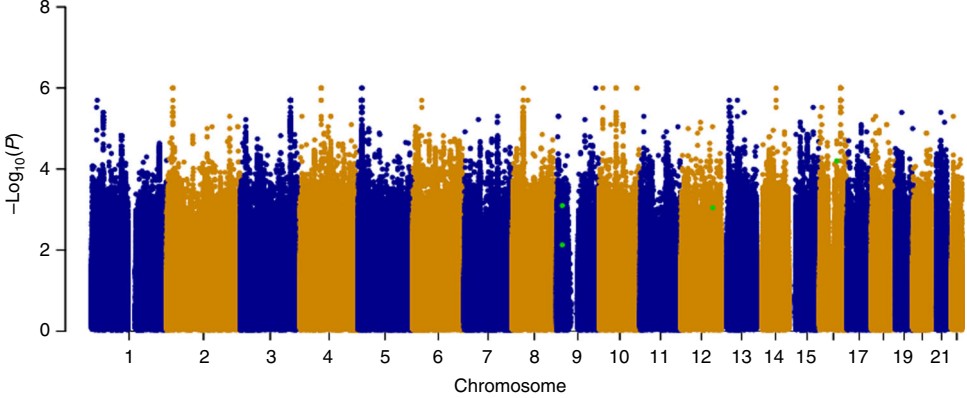

**Fig. 1** Genome-wide analysis of breast cancer survival. The Manhattan plot shows the result of the stage-1 meta-analysis. Results are plotted as −log$_{10}$ of the $P$-value from Cox regression. For each SNP the most significant $P$-value is selected from the analysis of either overall survival (OS) or disease-free survival (DFS) in all patients or the subset with early onset. The four most significant SNPs after meta-analysis of stages 1 and 2 are highlighted in green (rs410155 and rs12302097 associated with OS and DFS respectively in the whole cohort and rs715212 and rs10963755 associated with DFS in patients with early onset). This plot was produced using the qqman R package

(MAF=0.3) with modest effect sizes (HR=1.4 for all patients and HR=1.6 for early onset), no associations achieved genome-wide significance at stage-1. We therefore used our selection criteria (see Methods) to select 87 SNPs for assessment at stage-2, with $P$-values from the stage-1 meta-analysis ranging from $3.5 \times 10^{-7}$ to 0.008 (Supplementary Data 1).

To select additional SNPs, we conducted a literature search, which identified 73 variants associated with breast cancer survival, excluding studies that used any of our stage-1 cohorts. In our stage-1 meta-analyses, 57 of the published SNPs were genotyped and tested and 12 of them were replicated ($P_{meta} \le$ 0.05, Supplementary Data 2). From the 12 published SNPs with replication, 3 were selected for stage-2 analysis although 2 of these, which were the most significant, had already been chosen by our selection criteria. In total, 9 replicating SNPs were discounted either because nearby variants with higher significance had been selected ($n=3$), other variants in the gene had previously been tested ($n=2$) or because the significance level was modest ($P_{meta} \ge 0.01$, $n=4$). An additional 7 SNPs were selected for their association with onset age and or for their potential applications for risk prediction despite nonsignificance at stage-1, making a total of 95 SNPs for genotyping at stage-2 (Supplementary Data 1).

**Replication and meta-analysis of stages 1 and 2.** For replication at stage-2, 83 of the 95 selected SNPs were successfully genotyped by LGC Genomics using KASP chemistry in 1303 patients from the POSH cohort. QC excluded two SNPs with significant deviation from Hardy–Weinberg equilibrium ($P_{Hardy–Weinberg} \le 1 \times 10^{-10}$) and five monomorphic SNPs leaving 76 SNPs for analysis (Supplementary Data 1). After testing for association with OS and DFS, we identified three independent signals that were represented by four SNPs (1: rs715212 and rs10963755; 2: rs12302097; and 3: rs410155) with replication $P$-values from Cox regression ranging from 0.009 to 0.043 and effects in the same direction as the complimentary stage-1 analysis (Supplementary Data 1). For these SNPs, we used a fixed effects meta-analysis to determine their final effect size and significance by combining evidence from stages 1 and 2. Although none of the four SNPs reached a genome-wide level of significance, they were highly significant ($P_{meta}$ ranging from $3.54 \times 10^{-5}$ to $1.28 \times 10^{-4}$) and there was no evidence of heterogeneity between cohorts despite the reported differences in clinicopathologic features (Table 2).

Furthermore, the association with rs715212 was determined to be noteworthy by the FPRP $\le$ 0.2.

The effect size and significance of the replicating SNPs after meta-analysis of stages 1 and 2 and in each individual cohort (ABCFS, HEBCS, POSH stages 1 and 2 and SUCCESS-A) are shown in a forest plot (Fig. 2). The forest plot presents the most significant end point from the meta-analysis of stages 1 and 2 for all SNPs except rs12302097. For rs12302097, the association with DFS in all cases is presented rather than DFS in the early-onset subset, which is slightly more significant. This is because rs12302097 had similar effects in patients with early and late onset, indicating that this SNP influences prognosis in all patients.

rs715212 ($P_{meta} = 3.54 \times 10^{-5}$, HR=1.27) and rs10963755 ($P_{meta} = 3.91 \times 10^{-4}$, HR=1.22) were associated with an increased risk of disease progression, but interestingly this association was restricted to patients with early onset (Table 2). For both of these SNPs, the association with DFS was apparent in all four cohorts of early-onset patients (HEBCS, POSH1, SUCCESS-A and POSH2) and there was no evidence for association in patients with later onset (Fig. 2 and Table 2). The risk alleles for rs715212 ($P_{allelic\ association} = 1.03 \times 10^{-5}$, OR=1.36) and rs10963755 ($P_{allelic\ association} = 1.98 \times 10^{-4}$, OR=1.29) were significantly more common in early-onset patients with disease progression and there was no difference between patients with and without progression who were aged over 40 years old at diagnosis (Supplementary Table 2).

Although rs715212 and rs10963755 are only separated by 4.7 kb, the linkage disequilibrium between them is weak ($r^2 = 0.33$) and they were consequently treated as independent loci when selecting SNPs for replication. However, a multivariable analysis of DFS using pooled data in early-onset patients from HEBCS, POSH and SUCCESS-A showed that these SNPs represent a single association signal since rs10963755 did not retain significance ($P_{pooled} = 0.402$) after adjusting for rs715212 ($P_{pooled} = 0.003$).

rs12302097 is associated with an increased risk of relapse in all patients ($P_{meta} = 7.54 \times 10^{-5}$, HR=1.30) with a similar effect in patients with early ($P_{meta} = 6.77 \times 10^{-5}$, HR=1.45) and late onset ($P_{meta} = 0.0742$, HR=1.19, Table 2). Finally, rs410155 is associated with an increased risk of mortality in all cases ($P_{meta} = 1.28 \times 10^{-4}$, HR=1.34), with similar effect in patients with early ($P_{meta} = 0.0049$, HR=1.32) and late onset ($P_{meta} = 0.0066$, HR=1.32). Results from all survival and meta-analyses for the

**Table 2 Summary of the most significant SNPs from meta-analysis of stages 1 and 2 and their relationship with age of onset**

| End point | SNP[a] | Alleles[b] | MAF[c] | Flanking genes (distance to SNP) | All patients | | | Age of onset | | | | | | FPRP[d] |
| | | | | | | | | ≤ 40 | | | > 40 | | | |
| | | | | | $P_{meta}$ | HR (CI) | Q | $P_{meta}$ | HR (CI) | Q | $P_{meta}$ | HR (CI) | Q | |
|---|---|---|---|---|---|---|---|---|---|---|---|---|---|---|
| DFS | rs715212 | C/A | 0.270 | ADAMTSL1 | 0.0041 | 1.13 (1.04–1.23) | 0.214 | $3.54 \times 10^{-5}$ | 1.27 (1.13–1.42) | 0.773 | 0.7757 | 0.98 (0.87–1.11) | 0.960 | 0.188 |
| DFS | rs10963755 | C/G | 0.245 | ADAMTSL1 | 0.0147 | 1.11 (1.02–1.21) | 0.108 | $3.91 \times 10^{-4}$ | 1.22 (1.09–1.37) | 0.818 | 0.6834 | 0.97 (0.86–1.11) | 0.103 | 0.837 |
| DFS | rs12302097 | G/A | 0.061 | TXNRD1 (68.5 kb) CHST1 (38 kb) | $7.54 \times 10^{-5}$ | 1.30 (1.14–1.49) | 0.594 | $6.77 \times 10^{-5}$ | 1.45 (1.21–1.74) | 0.145 | 0.0742 | 1.19 (0.98–1.44) | 0.992 | 0.665 |
| OS | rs410155 | C/T | 0.062 | MT3 (8.7 kb) MT4 (11.6 kb) | $1.28 \times 10^{-4}$ | 1.34 (1.15–1.55) | 0.994 | 0.0049 | 1.32 (1.09–1.60) | 0.993 | 0.0066 | 1.32 (1.08–1.62) | 0.346 | 0.599 |

[a] The rs identifier from dbSNP
[b] Minor/major alleles
[c] Minor allele frequency (MAF) from 1000 genomes
[d] False positive report probability (FPRP)
$P_{meta}$, P-values from fixed effects meta-analysis; HR, hazard ratio; CI 95% confidence interval; Q, Cochran P-values for heterogeneity test

95 SNPs that were selected for genotyping at stage-2 are given in Supplementary Data 1.

Univariate Kaplan–Meier (KM) plots for the replicating SNPs and most significant outcomes were produced using pooled data from stages 1 and 2 (Fig. 3). Excluding rs410155, the significance levels and effect sizes from the pooled analyses were very similar to those obtained from meta-analysis (rs715212 $P_{pooled} = 1.94 \times 10^{-5}$, HR=1.28; rs10963755 $P_{pooled} = 6.48 \times 10^{-4}$, HR=1.21; rs12302097 $P_{pooled} = 1.65 \times 10^{-4}$, HR=1.28, Table 3). For rs410155, the pooled analysis was less significant and the HR was slightly smaller ($P_{pooled} = 0.015$, HR=1.20). The KM plots for rs715212 and rs12302097 were consistent with the additive model tested while the plots for rs10963755 and rs410155 were indicative of a dominant effect. For rs10963755 and rs410155, the dominant model was equally significant and the effect sizes were slightly larger (rs10963755 $P_{dominant} = 6.46 \times 10^{-4}$, HR=1.30; rs410155 $P_{dominant} = 0.0120$, HR=1.23).

**Multivariable models**. We used multivariable Cox regression in pooled data from stages 1 and 2 to determine whether the replicating SNPs were independent of the known prognostic factors that were available across all studies except ABCFS. After adjustment for ER status, tumour grade, maximum tumour diameter, axillary nodal status and study, the association between rs715212 and risk of disease progression in patients with early onset approached a genome-wide level of significance ($P_{multivariable} = 5.37 \times 10^{-8}$, HR=1.38, Table 3). The associations between rs10963755 and DFS in patients with early onset ($P_{multivariable} = 4.51 \times 10^{-5}$, HR=1.27) and between rs410155 and OS in all cases ($P_{multivariable} = 0.0023$, HR=1.28), also became stronger after adjustment for tumour characteristics and study compared with univariate analysis. For rs12302097, the association with DFS in all cases ($P_{multivariable} = 0.001$, HR=1.26) was less significant after adjustment.

**Functional inference**. To explore the functional relevance of the regions associated with survival, we used HaploReg v4.1[21], RegulomeDB[22] and SeattleSeq[23] to determine if the risk SNPs and their proxies ($r^2 \geq 0.2$) are located within putative functional elements such as active histone marks or transcription factor (TF) binding motifs (Supplementary Data 3). rs715212 and rs10963755 are located in intron 19 of the ADAMTSL1 gene and both are reported to alter TF binding motifs (Fig. 4a and Supplementary Data 3). The chromatin surrounding rs715212 is characterized as an enhancer in breast myoepithelial primary cells (strongly enriched for TF binding sites, moderately enriched for

DNase peaks and conserved elements), while rs10963755 maps to a quiescent region. rs12302097 is located in a 64 kb region of linkage disequilibrium (LD) ($r^2 > 0.2$) between *carbohydrate sulfotransferase 11* (CHST11, 38 kb upstream) and *thioredoxin reductase 1* (TXNRD1, 68.5 kb downstream). The chromatin in this region has properties of weak transcription in variant human mammary epithelial cells (vHMECs) and the SNP has been shown to alter several TF binding motifs (Fig. 4b and Supplementary Data 3). Finally, rs410155 is located between two metallothionein genes: *MT3* (8.7 kb upstream) and *MT4* (11.6 kb downstream, Fig. 4c). Although the surrounding chromatin in breast myoepithelial primary cells is quiescent, this SNP has also been predicted to alter TF binding motifs.

To gain further insight into the functional basis of each risk SNP and its linked SNPs ($r^2 \geq 0.2$), we used the Genotype-Tissue Expression (GTEx)[24] portal to perform expression quantitative trait locus (eQTL) analysis of the genes at either side of the index SNP in breast mammary tissue. Although none of the index SNPs were associated with the expression of their flanking genes, SNPs in weak LD with rs12302097 were found to be nominally associated with the expression of *CHST11* (rs56372209, $P_{FastQTL} = 0.041$, $r^2 = 0.3$) and *TXNRD1* (rs73183724, $P_{FastQTL} = 0.031$, $r^2 = 0.25$, Supplementary Data 3). Using GTEx we also found that several SNPs in complete LD with rs410155 including rs381706 were associated with the expression of MT1E ($P_{FastQTL} = 0.044$) and *MT1F* ($P_{FastQTL} = 0.026$) in breast mammary tissue (Supplementary Data 3). In whole blood, rs410155 is associated with the expression of *MT1E* ($P = 2.04 \times 10^{-4}$) and *MT1F* ($P = 2.22 \times 10^{-6}$) and rs12302097 is associated with *TXNRD1* expression ($P = 0.001$)[25].

Several genes related to ADAMTSL1 have been implicated in the development of normal breast tissue and in the initiation and progression of breast cancer including ADAMTS1[26], ADAM10[27], ADAM12[28] and ADAM17[29]. Of these, ADAM17, which plays a key role in normal breast development via its cleavage and release of amphiregulin (AREG) from the surface of breast epithelial cells, appeared to be particularly relevant because two independent studies have shown that *AREG* is overexpressed in ER-positive breast tumours from premenopausal women vs. postmenopausal women[17,18]. We therefore used GTEx to perform further eQTL analysis, and found that rs715212 is nominally associated with the expression of *AREG* ($P_{FastQTL} = 0.035$) in breast mammary tissue (Supplementary Fig. 6).

**Triple-negative breast cancer**. TNBC is an aggressive breast cancer subtype with limited treatment options due to the lack of

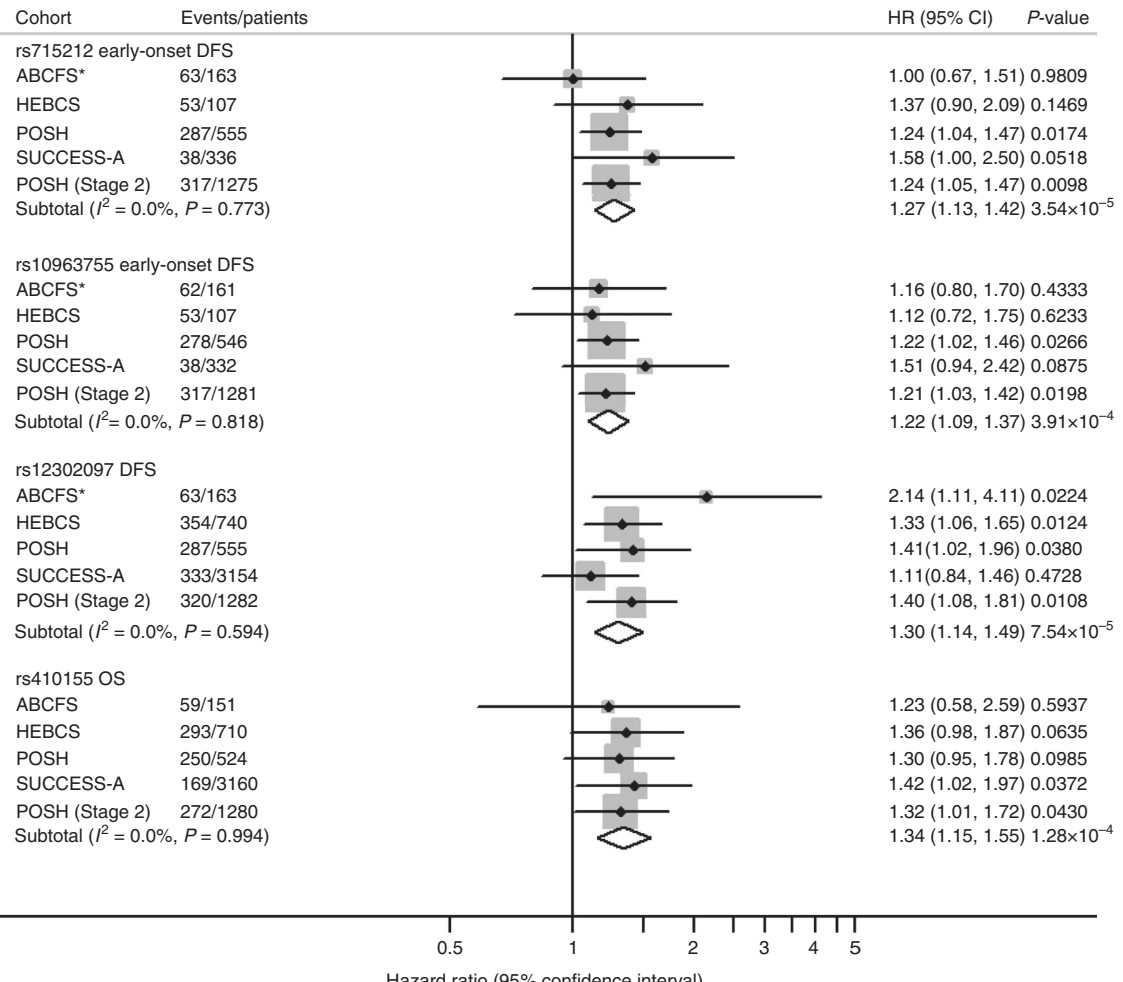

**Fig. 2** Forest plot and meta-analysis for the four most significant SNPs associated with overall survival (OS) or disease-free survival (DFS). Forest plot showing the event rate, hazard ratio (HR), 95% confidence interval (CI) and significance level (P-value) from Cox regression in each cohort and the combined analysis for the most significant SNPs associated with DFS and OS. ABCFS*: evidence for association with OS in the ABCFS cohort is shown for each SNP but these results are excluded from the meta-analyses of DFS. The SNP subtotal rows show the result for a fixed effects meta-analysis across four studies for rs715212, rs10963755 and rs12302097 and five studies for rs410155 using $I^2$ and Cochran Q-statistic to assess heterogeneity in effect sizes between cohorts

expression of ER, PR and HER2 receptors. Studies have shown that the immunoreactivity of *MT3*, which flanks rs410155, is associated with poor prognosis in TNBC patients[30,31]. To explore the relationship with TNBC, the survival analyses were repeated in a pooled subset of TNBC patients. rs410155 was the only SNP that had a stronger association with prognosis in patients with TNBC ($P_{pooled} = 0.004$ for OS in TNBC vs. $P_{pooled} = 0.014$ for OS in all cases, Supplementary Table 3 and Supplementary Fig. 7).

## Discussion

A recent study highlighted the difficulties of detecting variants linked to breast cancer survival that were predicted to have small effect sizes[10]. They suggested that very large sample sizes will be required to detect common variants conferring HRs that may be as low as 1.1, which is in line with findings from GWASs of breast cancer risk. To our knowledge, only two loci, rs4458204[32] and rs2059614[33], have confidently been associated with breast cancer survival at genome-wide levels of significance despite the analysis of up to 37,954 patients and an event rate of 7.6% ($n = 2900$ deaths). Our hypothesis was that enrichment for well-characterized early-onset cases with worse prognosis might facilitate detection by increasing the event rate, reducing genetic

heterogeneity and because variants with larger effect sizes might underlie more aggressive forms of disease. Although the present study has a comparatively small sample size ($n = 6042$ patients including 2315 aged $\leq 40$ years at diagnosis), the event rate is significantly higher over all patients (OS 18.2%, DFS 22.5%) and the early-onset subset (OS 26.1%, DFS 30.9%) which is one of the main determinants of power. Consequently, the combined analysis of stages 1 and 2 for DFS is estimated to have 80% power to detect common SNPs (MAF=0.3) with an effect size of HR=1.31 in all patients and HR=1.42 in the early-onset subset at a genome-wide level of significance.

Despite sufficient statistical power to detect common SNPs with modest effect sizes no variants were identified at a genome-wide level of significance. This may reflect the confounding impacts of a large number of factors that influence survival times, including phenotypic heterogeneity, tumour biology and treatment. However, three signals had suggestive levels of significance (1: rs715212 and rs10963755; 2: rs12302097; and 3: rs410155) without heterogeneity between cohorts including one that was estimated to be noteworthy according to the FPRP (rs715212 FPRP< 0.2). With the exception of rs12302097, the effect size (HR) and significance of these associations became stronger after

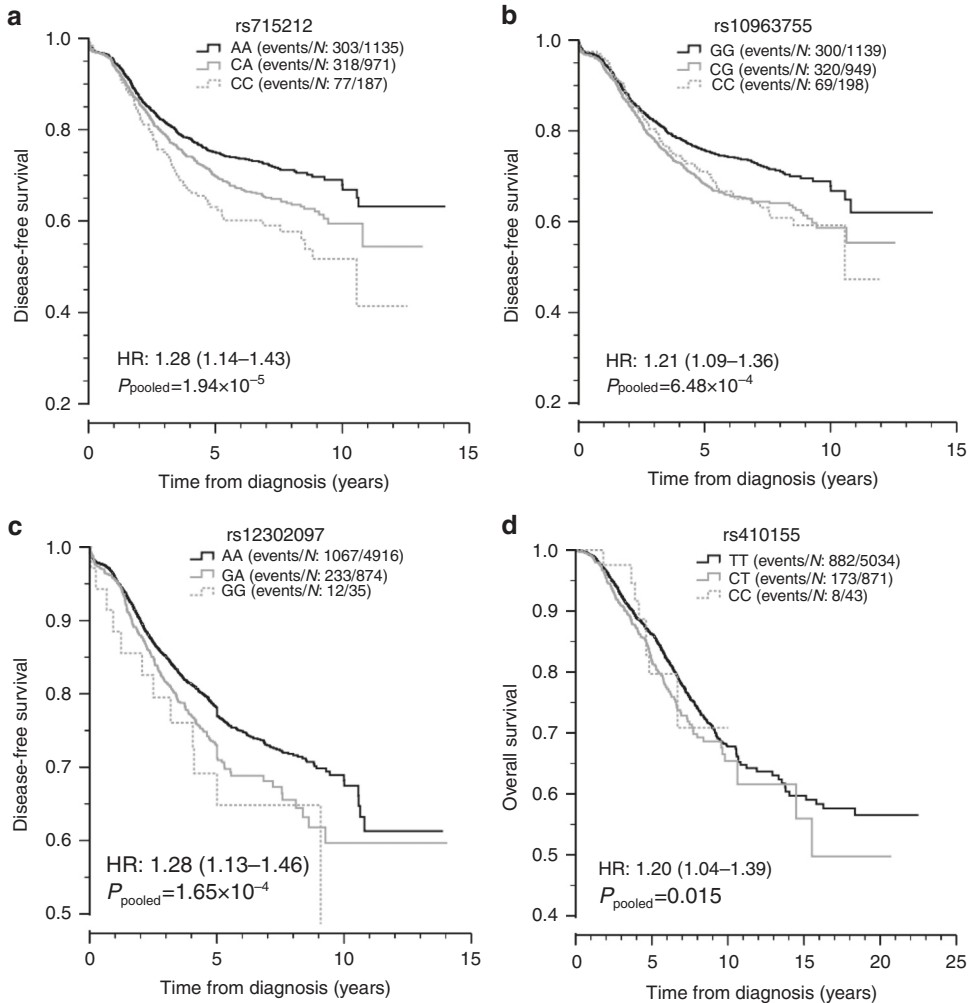

**Fig. 3** Kaplan–Meir survival plots for the four most significant SNPs identified by meta-analyses. Kaplan–Meier plots from univariate analysis of the most significant SNP associated with disease-free survival (DFS) in cases with early onset (**a**, rs715212 and **b**, rs10963755), DFS in all cases (**c**, rs12302097) and overall survival (OS) in all cases (**d**, rs410155). For OS, the data from all five cohorts (ABCFS, HEBCS, POSH stages 1 and 2 and SUCCESS-A) was pooled whereas for DFS data were pooled across four cohorts because DFS was not recorded in the ABCFS cohort. HR: hazard ratio with 95% confidence interval

adjustment for tumour characteristics in multivariable models. This suggests that tumour characteristics are confounding factors and that accounting for them in a multivariable model will increase the accuracy of effect size estimates. Previous studies have shown that similar adjustments for known prognostic factors increases statistical power in the analysis of time-to-event outcomes[34].

The signal involving rs715212 and rs10963755 was associated exclusively with disease progression in patients with early onset and, in a pooled analysis with adjustment for tumour characteristics and study, rs715212 approached a genome-wide level of significance. However, for both SNPs there was no difference in risk allele frequency between patients with early and late onset. This suggests that the association may involve an interaction with another factor that is unique to early-onset patients such as the expression or somatic mutation profile of the tumour.

Both rs715212 and rs10963755 are located in intron 19 of the *ADAMTSL1* gene which encodes a secreted glycoprotein and is a member of the *ADAMTS* (a disintegrin and metalloproteinase with thrombospondin motif) family (Fig. 4a). Previous studies have shown that ADAMTSL1 is a component of the extracellular matrix that may function in cell–cell or cell–matrix interactions or may regulate other *ADAMTS* proteases[35]. Although *ADAMTSL1* is primarily expressed in skeletal muscle, it has been

seen in other tissues including breast and methylation studies have shown that it is hypermethylated in ER-positive breast cancer tumours[36,37].

We have shown that rs715212 is nominally associated with the expression of AREG and that the chromatin surrounding rs715212 is predicted to be functional. These findings provide tentative evidence that rs715212 and/or linked variants may influence disease progression in early-onset patients by altering the methylation or functionality of *ADAMTSL1* and/or the expression of *AREG* either directly or via regulation of other members of the *ADAMTS* gene family. However, it is important to stress that further functional studies are required to verify the association between rs715212 and *AREG* expression and determine the biological mechanism.

Of the two remaining SNPs with moderate association, rs12302097 is associated with disease progression in all patients and rs410155 is associated with OS in all patients. For rs12302097, we have shown that SNPs in weak LD are associated with the expression of both flanking genes (*CHST11* and *TXNRD1*) and both of these genes have previously been associated with breast cancer[38–41]. This suggests that *CHST11* and or *TXNRD1* may influence prognosis.

The final variant, rs410155, is located between two metallothionein genes, *MT3* and *MT4*. While *MT4* has not been

**Table 3 Univariable and multivariable analysis of pooled data**

| Covariates at diagnosis | Univariable | | | Multivariable | |
|---|---|---|---|---|---|
| | HR (95% CI) | $P_{univariable}$ | Events/cases | HR (95% CI) | $P_{multivariable}$ |
| *Model 1 (DFS early onset)* | | | | | |
| rs715212 | 1.28 (1.14–1.43) | $1.94 \times 10^{-5}$ | 698/2293 | 1.38 (1.23–1.56) | $5.37 \times 10^{-8}$ |
| ER status | 0.78 (0.67–0.91) | 0.001 | 702/2295 | 1.09 (0.90–1.31) | 0.377 |
| Grade | 1.53 (1.33–1.75) | $1.92 \times 10^{-9}$ | 684/2260 | 1.37 (1.17–1.61) | $9.05 \times 10^{-5}$ |
| Tumour size (mm) | 1.02 (1.01–1.02) | $1.00 \times 10^{-13}$ | 667/2258 | 1.01 (1.01–1.02) | $1.01 \times 10^{-13}$ |
| Nodal status | 2.54 (2.16–3.00) | $1.00 \times 10^{-13}$ | 676/2271 | 2.37 (1.98–2.83) | $1.00 \times 10^{-13}$ |
| Cohort | 0.63 (0.59–0.67) | $1.00 \times 10^{-13}$ | 705/2315 | 0.62 (0.58–0.67) | $1.00 \times 10^{-13}$ |
| *Model 2 (DFS early onset)* | | | | | |
| rs10963755 | 1.21 (1.09–1.36) | 0.00065 | 689/2286 | 1.27 (1.13–1.43) | $4.51 \times 10^{-5}$ |
| ER status | 0.78 (0.67–0.91) | 0.00117 | 702/2295 | 1.10 (0.92–1.33) | 0.301 |
| Grade | 1.53 (1.33–1.75) | $1.92 \times 10^{-9}$ | 684/2260 | 1.38 (1.18–1.62) | $7.33 \times 10^{-5}$ |
| Tumour size (mm) | 1.02 (1.01–1.02) | $1.00 \times 10^{-13}$ | 667/2258 | 1.01 (1.01–1.02) | $1.00 \times 10^{-13}$ |
| Nodal status | 2.54 (2.16–3.00) | $1.00 \times 10^{-13}$ | 676/2271 | 2.31 (1.93–2.77) | $1.00 \times 10^{-13}$ |
| Cohort | 0.63 (0.59–0.67) | $1.00 \times 10^{-13}$ | 705/2315 | 0.63 (0.58–0.68) | $1.00 \times 10^{-13}$ |
| *Model 3 (DFS all cases)* | | | | | |
| rs12302097 | 1.28 (1.13–1.46) | $1.65 \times 10^{-4}$ | 1312/5825 | 1.26 (1.10–1.44) | 0.001 |
| ER status | 0.66 (0.59–0.74) | $7.11 \times 10^{-13}$ | 1298/5756 | 0.80 (0.70–0.91) | 0.001 |
| Grade | 1.50 (1.37–1.65) | $1.00 \times 10^{-13}$ | 1261/5695 | 1.56 (1.40–1.73) | $1.01 \times 10^{-13}$ |
| Tumour size (mm) | 1.02 (1.02–1.02) | $1.00 \times 10^{-13}$ | 1256/5711 | 1.02 (1.01–1.02) | $1.00 \times 10^{-13}$ |
| Nodal status | 1.98 (1.75–2.24) | $1.00 \times 10^{-13}$ | 1274/5747 | 2.10 (1.84–2.41) | $1.00 \times 10^{-13}$ |
| Cohort | 0.61 (0.58–0.64) | $1.00 \times 10^{-13}$ | 1316/5840 | 0.58 (0.55–0.62) | $1.00 \times 10^{-13}$ |
| *Model 4 (OS all cases)* | | | | | |
| rs410155 | 1.20 (1.04–1.39) | 0.0154 | 1063/5945 | 1.28 (1.09–1.51) | 0.0023 |
| ER status | 0.60 (0.53–0.68) | $1.00 \times 10^{-13}$ | 1081/5919 | 0.66 (0.57–0.77) | $5.92 \times 10^{-8}$ |
| Grade | 1.72 (1.54–1.91) | $1.00 \times 10^{-13}$ | 990/5692 | 1.65 (1.46–1.87) | $1.06 \times 10^{-13}$ |
| Tumour size (mm) | 1.02 (1.02–1.02) | $1.00 \times 10^{-13}$ | 981/5708 | 1.02 (1.02–1.02) | $1.00 \times 10^{-13}$ |
| Nodal status | 2.12 (1.85–2.44) | $1.00 \times 10^{-13}$ | 998/5744 | 2.22 (1.90–2.59) | $1.00 \times 10^{-13}$ |
| Cohort | 0.79 (0.76–0.83) | $1.00 \times 10^{-13}$ | 1101/6039 | 0.67 (0.63–0.71) | $1.00 \times 10^{-13}$ |

Univariable and multivariable analyses were performed with pooled data from ABCFS, HEBCS, POSH (stages 1 and 2) and SUCCESS-A for overall survival (OS). The ABCFS cohort was excluded from the analysis of disease-free survival (DFS). For univariate analyses, the number of events and cases is slightly different for each covariate due to variation in the number of cases with missing data. For multivariable analyses, the following number of events and cases were used: model 1: 642/2172, model 2: 634/2166, model 3: 1201/5538, model 4: 911/5482. In each model the hazard ratios (HRs) and 95% confidence interval (CI) were adjusted for oestrogen receptor status (ER), grade, maximum tumour size, nodal status, SNP and cohort. $P_{univariable}$, P-values from univariable Cox regression; $P_{multivariable}$, P-values from multivariable Cox regression.

associated with breast cancer, *MT3* is overexpressed in breast cancer cells, which is associated with increased invasiveness and higher concentrations of matrix metallopeptidase 3[30,31]. These findings suggest that elevated expression of *MT3* may underlie the association that we have identified between overall survival and genetic variation at rs410155.

The genome-wide significant SNPs identified by previous studies were both associated with survival in ER-negative patients[32,33]. We therefore repeated the stage-1 meta-analysis using ER-negative patients only ($n = 1637$) but failed to replicate the findings for rs4458204 ($P_{meta} = 0.771$) and rs2059614 ($P_{meta} = 0.482$) despite a small overlap in the patients tested by these studies ($n = 196$ from HEBCS for rs2059614 and $n = 315$ from POSH for rs4458204). However, the current study lacks power to replicate these findings given the small number of ER-negative patients and low MAF for rs4458204 (MAF=0.12) and rs2059614 (MAF=0.03).

In a subsequent study which aimed to replicate SNPs with suggestive significance levels using the same patient cohorts, Pirie et al.[10] identified 12 variants with nominal significance ($P < 0.05$) including 7 that were associated with ER-positive disease. Eleven of these SNPs were genotyped and tested by the current study which replicated the association with rs1800566 in all patients ($P_{meta} = 0.01$) and rs10477313 in ER-positive patients ($P_{meta} = 0.013$). The nine remaining variants showed no evidence of association, although five of these had MAF < 0.1.

The current study has several limitations which must be noted. First, no variants were identified at a genome-wide level of significance and the most significant results were derived from a

multivariable analysis which adjusted for the confounding effect of tumour characteristics. Second, none of the survival analyses were adjusted for treatment and the ABCFS cohort could not be included in the analyses of DFS, early onset and multivariable models because these variables were unavailable. Third, although the study is well powered to detect common SNPs we estimate that it only has 20% power to detect rare SNPs (MAF=0.1) associated with DFS with HRs of 1.31 for all patients and 1.44 for those with early onset. Fourth, the associations with gene expression involve healthy participants and have relatively modest significance levels. Further analysis with larger sample sizes and adjustment for additional clinicopathological factors including treatment (chemotherapy and hormone therapy) may provide more information that could further improve survival analysis. Further functional studies involving breast cancer patients and including epigenetic mechanisms should be performed to provide more insights about the three association signals identified in the present study.

Our meta-analysis identified three independent signals associated with breast cancer prognosis that are independent of the classical prognostic factors. Interestingly, the signal located in *ADAMTSL1* was only associated with disease progression in patients with early onset. This suggests that unique disease mechanisms may influence survival in younger women and provide some biological insight into why younger-onset breast cancer has a worse prognosis. We have also discussed the possible impact of these variants on the methylation of *ADAMTSL1*, its interaction with other members of the *ADAMTS* gene family and their association with the expression of other biologically relevant

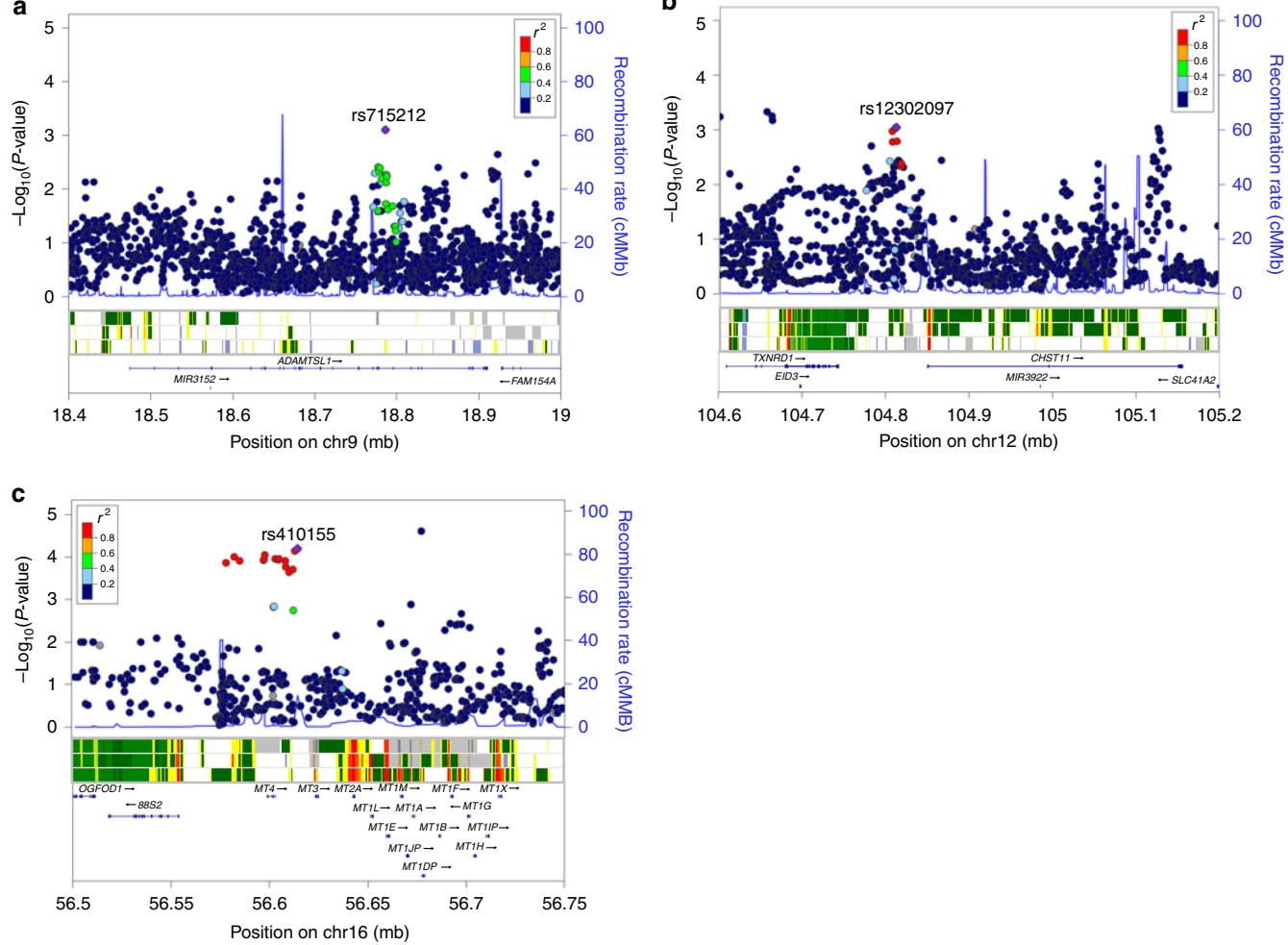

**Fig. 4** Regional plots of association with survival (OS or DFS) at stage-1 meta-analysis, recombination rate and gene context for the most significant SNPs. Results from the stage-1 meta-analyses in a region surrounding the most significant SNP associated with DFS in patients with early onset (**a**, rs715212 and rs10963755), DFS in all patients (**b**, rs12302097) and OS in all patients (**c**, rs410155). In each plot, a purple diamond identifies the index SNP and the colour of other SNPs represent their linkage disequilibrium ($r^2$) with the index SNP from light blue ($r^2 \leq 0.4$) to red ($r^2 \geq 0.8$). The middle panel displays the 15 state chromatin segmentation track (ChromHMM) in breast variant human mammary epithelial cells (vHMECs, E028), mammary epithelial primary cells (HMECs, E119) and breast myoepithelial primary cells (E027) using data from the HapMap ENCODE Project. The lower panels show genes and their direction of transcription (arrows). Physical positions are relative to build 37 (hg19) of the human genome

genes such as *AREG* in the context of breast cancer prognosis. The SNPs identified in this study have the potential to improve the accuracy of prognostic estimates and stratification of patients into treatment groups. Moreover, the gene implicated by these SNPs may warrant further investigation as novel therapeutic targets and some are already under investigation for this purpose.

## Methods

**Patient cohorts for discovery and replication stages**. At stage-1, breast cancer samples were selected from four cohorts: (1) ABCFS (http://epi.unimelb.edu.au/research/cancer/breast/)[42], (2) HEBCS[43], (3) POSH from the United Kingdom[44] and (4) a prospective randomized phase III clinical trial comparing FEC-Doc Chemotherapy vs. FEC DocG chemotherapy (SUCCESS-A) from Germany[45]. Prior to QC, these data sets consisted of: 214 incident cases in ABCFS diagnosed between 1992 and 1995 with a first primary invasive breast cancer before the age of 40 years and living in Melbourne or Sydney; 832 breast cancer patients in HEBCS aged 22–96 years and treated in the Helsinki region between 1997 and 2004; 574 patients from the POSH study aged ≤ 40 years at diagnosis of invasive breast cancer between 2000 and 2007; and 3277 patients from the SUCCESS-A trial diagnosed with primary epithelial invasive carcinoma and recruited between 2005 and 2007 (Supplementary Table 1). All patients gave informed consent and the studies were approved by the relevant ethics committees[42–45]. Histopathological and survival data were collected for all patients. In HEBCS, 590 patients were unselected and

242 were familial[43]. In the POSH cohort, 274 patients were genotyped as part of a larger study on TNBC in which there is little or no tumour expression for ER, PR and human epidermal growth factor receptor 2 (HER2)[46]. To increase power, the remaining samples from the POSH cohort were enriched for survival extremes corresponding to patients with early distant metastasis or death ($n = 193$, median OS=2.8 years) and patients with long-term event free survival (n = 96, median OS=8.7 years).

A further 1303 patients from the POSH cohort[44] who were unselected for any survival differences and were independent of the stage-1 data set were used for replication analysis at stage-2 (Supplementary Table 1). These patients were aged ≤ 40 years at diagnosis of invasive breast cancer and had self-reported ethnicities of White/Caucasian ($n = 1285$), eastern European ($n = 3$), Greek ($n = 2$), South African ($n = 2$) and Jewish ($n = 11$).

To assess the similarity of clinicopathologic features between all cohorts with data ($n = 3$ to 5), Pearson's $\chi^2$ test was used for categorical traits and Kruskal–Wallis rank sum tests were used for continuous traits (Table 1).

**Genotyping and QC at stage-1**. At stage-1, samples were genotyped using the Illumina 610k array for ABCFS, Illumina 550K array for HEBCS[47], Illumina 660-Quad array for POSH by The Mayo Clinic (Rochester, MN, USA) and Genome Institute of Singapore (GIS)[48] and the HumanOmniExpress-FFPE BeadChip for SUCCESS-A. Standard QC measures were applied to the genotypic data from stage-1, which removed SNPs with MAFs of < 5%, SNPs and individuals with > 10% missing genotypes and SNPs with significant deviations from Hardy–Weinberg equilibrium ($P$-value ≤ $1 \times 10^{-10}$). Strand issues, where the allele

coding differs between cohorts, were resolved by combining the cohorts and flipping strands for SNPs with more than two alleles and by ensuring that the MAFs were similar between cohorts particularly for A/T and G/C SNPs where strand issues cannot be detected by allelic excess. The reported gender of the individuals was verified against that predicted from their genotypic data. All duplicate samples, individuals with incomplete phenotypic data for survival analyses and samples that were cryptically related (pairwise-identity by state > 86%) were excluded from the stage-1 cohorts. Samples that were inferred to have non-European ancestry by multidimensional scaling analysis against reference populations from HapMap were excluded so that the remaining samples in each cohort formed a single cluster which overlapped with the Caucasian reference (CEU, Supplementary Fig. 1). Across the four stage-1 cohorts, a total of 143,380 SNPs and 158 patients were removed during these QC steps leaving 4739 patients and observed genotypes at 813,964 SNPs for analysis. The number of SNPs and patients removed and remaining in each cohort is shown in Supplementary Table 1. All QC procedures were carried out using PLINK[49].

**Imputation of the stage-1 data and further QC**. To aid meta-analysis and to increase the resolution of the stage-1 data, additional SNPs were imputed using MACH 1.0 (http://www.sph.umich.edu/csg/abecasis/MACH/index.html). The reference data for imputation were SNP genotypes from HapMap phase 2 and phased haplotypes from the CEPH population (Utah residents with ancestry from northern and western Europe, CEU). The imputed genotypes were quality controlled by excluding SNPs with: a posterior probability < 0.9, MAF <5%, >10% missing genotypes or significant deviations from Hardy–Weinberg equilibrium (P-value $\leq 1 \times 10^{-10}$). Following these QC steps, 5,848,861 imputed SNPs across the four stage-1 cohorts remained for analysis (Supplementary Table 1).

**Power calculations**. The power to detect SNPs associated with OS and DFS in all cases and patients with early-onset breast cancer in the combined stage-1 cohorts was estimated using the survSNP program in R[50] with an additive genetic risk model and type 1 error rate (α) of $5 \times 10^{-8}$ (Supplementary Fig. 2). A range of modest genotype HRs (1.1–2.0) and risk allele frequencies (0.05–0.3) were used along with the documented values for sample size and event rate after QC (Supplementary Table 1).

**Cox regression**. To identify SNPs influencing prognosis we used the formetascore command in GenABEL[51] to perform Cox regression analyses of OS and DFS, with correction for ER status which is the only variable that is recorded in all four cohorts, has the most complete data and is associated with survival[52]. SNPs were coded according to the number of rare alleles (0–2). For OS, follow-up times were defined as the duration between breast cancer diagnosis and death from any cause or last follow-up if alive. For DFS, the follow-up times were defined as the time between diagnosis and disease progression in the form of local recurrence, distant metastasis or death from any cause, whichever occurred first, or last follow-up if alive. Patients who had not experienced an event at the time of analysis were censored at their date of last follow-up.

To identify variants associated with early-onset breast cancer, the OS and DFS analyses were repeated in a subset of 2315 patients who were aged ≤ 40 years at diagnosis from HEBCS ($n = 119$), POSH stage-1 ($n = 556$), POSH stage-2 ($n = 1303$) and SUCCESS-A ($n = 337$). For the early-onset analysis, the POSH cohort was particularly important because all of the POSH patients were aged ≤ 40 years at diagnosis. The ABCFS data set did not contain data on age of onset, local recurrence or distant metastasis and therefore could not be used for the analysis of DFS or early onset. For SNPs associated with early onset, the relationship between prognosis and onset age was further explored by: (1) repeating the OS and DFS survival analyses in a subset of patients from HEBCS ($n = 679$) and SUCCESS-A ($n = 2846$) that were aged > 40 years at diagnosis and (2) testing for allelic association with OS and DFS events in patients with early (aged ≤40 years at diagnosis) and late onset (aged > 40 years at diagnosis) using Pearson's $\chi^2$ test.

The mean difference in time between age at diagnosis and age at registration was 0.78 years (s.d.=1.16 years) over all cohorts.

**Visualization of stage-1 results**. To verify the robustness of our QC measures and to examine the possibility of confounding factors such as population stratification, we used the qqnorm and qqplot procedures in R[53] to generate QQ plots of the observed and expected P-values under the null distribution in the stage-1 data (Supplementary Figs. 3–5).

To visualise the stage-1 results and SNPs selected for follow-up, the qqman package in R[54] was used to generate a Manhattan plot and highlight the most significant SNPs associated with OS and DFS (Fig. 1). For SNPs with significant replication, Locus Zoom[55] was used to generate regional plots of the stage-1 data to show the pattern of association surrounding the index SNP with respect to LD with neighbouring SNPs, underlying recombination rate and gene context (Fig. 4).

**Meta-analysis**. To select SNPs for follow-up at stage-2, we used PLINK[49] to perform a fixed effects inverse variance-weighted meta-analysis of the stage-1 results for OS and DFS in all cases and the subset of patients with early onset. A fixed effects meta-analysis was used under the assumption that SNPs have one true

effect size and that any differences between studies were most likely to be due to sampling variation. To estimate heterogeneity in effect size between studies we used the $\chi^2$-based Cochran Q-statistic and $I^2$ which gives the percentage of variation across studies that is due to heterogeneity rather than chance[56]. The same methodology was used to determine the final significance and effect size of SNPs by meta-analysis of the Cox regression results from stages 1 and 2. To visualise the meta-analysis, we produced a forest plot using Stata version 12 (Fig. 2)[57]. Following meta-analysis of stages 1 and 2, data from all cohorts were pooled and KM plots were generated for the most significant SNPs using Stata version 12[57].

**False positive report probability**. To assess the reliability of the associations from meta-analysis of stages 1 and 2 we calculated the FPRP which describes the probability of no true association between a genetic variant and disease, given a statistically significant finding[58]. The FPRP was calculated using a low prior probability of 0.0001, which is expected for a random SNP, to detect a hazard ratio of 1.3. A threshold of FPRP ≤ 0.2 was used to identify noteworthy associations.

**Selection, genotyping and QC of SNPs at stage-2**. Completely unbiased methods of SNP selection have no means of excluding false positives which are likely to be among the most significant signals. They will also neglect moderately significant SNPs in favour of the most significant SNPs despite potentially overwhelming support from correlated SNPs and proximity to biologically relevant genes. To select the most promising SNPs for follow-up, we therefore used a clumping procedure in PLINK[49] to generate a shortlist of index SNPs with support from correlated SNPs (SNPs $r^2 \geq 0.5$, within 500 kb). Priority, but not exclusivity, was then given to index SNPs that were close to a relevant gene according to annotation from GeneAlacart (https://genealacart.genecards.org/). Two shortlists of index SNPs were made which used either a stringent (index SNP $P_{meta} \leq 0.001$ and correlated SNP $P_{meta} \leq 0.01$) or moderate set of P-value thresholds (index SNP $P_{meta} \leq 0.01$ and correlated SNP $P_{meta} \leq 0.1$). SNPs were selected from the stringent shortlist first ($n = 50$) and then from the moderate shortlist ($n = 37$). Since priority but not exclusivity was given to SNPs close to relevant genes, 20 SNPs were selected on a completely unbiased basis and 67 were selected from the unbiased shortlist because they were close to a relevant gene (Supplementary Data 1).

To select additional SNPs, we performed a literature search, which identified 73 variants that have previously been associated with breast cancer survival (OS, DFS or breast cancer-specific survival) in independent cohorts. These published SNPs were cross-referenced with our stage-1 meta-analysis and 8 additional SNPs were selected on the basis of their published association with onset age and/or because the gene implicated had potential applications for diagnosis, risk prediction or therapeutic intervention.

The 95 SNPs selected for replication were genotyped by LGC Genomics (Hoddeson, UK) in 1303 patients from the POSH cohort. The genotypes were quality controlled by excluding SNPs with >10% duplicate error rate, >10% missing genotypes or significant deviations from Hardy–Weinberg equilibrium (P-value $\leq 1 \times 10^{-10}$).

**Multivariable Cox regression**. We used multivariable Cox regression in pooled data from stages 1 and 2 to determine whether SNPs with the most significant impacts on survival were independent of the known prognostic factors that were available across all studies. Data on ER status (negative=0, positive=1) tumour grade (1 to 3), maximum tumour diameter (mean 25.7, range 0 to 220 mm) and axillary nodal status (not affected=0, affected=1) were available for 93% of the cases that passed QC ($n = 5622/6042$). These prognostic factors along with the cohort used were treated as covariates and were entered into the proportional hazards model in order of significance. All survival analyses were carried out using Stata version 12 and the P-values reported were two-sided at 5% significance.

**Functional annotation of significant SNPs**. To explore the functional relevance of the regions associated with survival, we used HaploReg v4.1[21], RegulomeDB[22] and SeattleSeq[23] to interrogate ENCODE data[59] and annotate the risk SNPs and their linked SNPs ($r^2 \geq 0.2$) with respect to: histone modifications, DNAseI hypersensitivity, proteins bound, disruption of regulatory motifs, conservation metrics from genomic evolutionary rate profiling (GERP)[60] and combined annotation-dependent depletion scores (CADD)[61], and functionality scores from RegulomeDB. The scores from RegulomeDB were generated using data from Gene Expression Omnibus (GEO), ENCODE and published literature. Variants with a RegulomeDB score of 3 are likely to affect binding while variants scoring 4–6 have minimal evidence for functional activity.

Additionally, candidate regions were annotated with 15 state chromatin segmentation in breast vHMECs (E028), mammary epithelial primary cells (HMEC, E119) and breast myoepithelial primary cells (E027). These chromatin states categorize noncoding DNA into functional regulatory elements such as enhancers and quiescent regions that are respectively enriched and depleted for phenotype-associated SNPs[62]. The chromatin states were generated by computational integration of binarized chromatin immunoprecipitation sequencing data using a multivariable Hhidden Markov model that explicitly models the combinatorial patterns of observed modifications[63].

To gain further functional insight, eQTL analysis was performed for all SNPs in LD ($r^2 \geq 0.2$) with the index SNPs using the GTEx portal (V6, dbGaP Accession phs000424.v6.p1)[24] to query RNAseq data from breast mammary tissue in 183 samples with genotype data.

**Data availability**. All relevant summary statistics from the ABCFS, HEBCS and POSH cohorts are available from the authors for collaborative research upon request to the corresponding author. The SUCCESS-A data are available via authorized access from dbGaP (Study Accession: phs000547.v1.p1).

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

## Acknowledgements
W.T., D.E., A.C. and L.K. were supported by Breast Cancer Now. L.K. was also supported by the Faculty of Natural and Environmental Sciences, University of Southampton. The Australian Breast Cancer Family Study (ABCFS) and J.L.H. were supported by the National Health and Medical Research Council (NHMRC), Victorian Health Promotion Foundation, Cancer Council New South Wales, Cancer Council Victoria, Cancer Australia and National Breast Cancer Foundation. The Helsinki Breast Cancer Study (HEBCS) thanks Dr. Kristiina Aittomäki and research nurse Irja Erkkilä for their help with collecting patient data and samples. HEBCS has been supported by the Helsinki University Central Hospital Research Fund, the Academy of Finland (266528), the Sigrid Juselius Foundation and the Cancer Society of Finland. Work in Singapore by J.L. was funded by the Agency for Science, Technology and Research (A*STAR). Work in America by F.J.C was funded by The Breast Cancer Research Foundation and a Specialized Program of Research Excellence (SPORE) in Breast Cancer to Mayo Clinic (P50 CA116201).

## Author contributions
The study was designed and coordinated by W.T., D.E. and A.C. W.T. and L.K. performed statistical analysis. W.T., D.E., A.C. and L.K drafted the manuscript. SNP array and clinical data were provided by J.L.H. for the ABCFS cohort, H.N., S.K. and C.B. for the HEBCS cohort, P.A.F., B.R. and W.J. for the SUCCESS-A study. SNP array genotypes for subsets of the POSH cohort were provided by J.L and F.J.C. Clinical data for the POSH cohort were provided by T.M., L.D. and S.G.

## Additional information

**Competing interests:** The authors declare no competing financial interests.

