## [Peer Review File · Nature Communications]

Reviewers' comments:

Reviewer #1 (Expert in cancer genomics):

Kadalayil et al. investigated SNPs associated with breast cancer prognosis using a two-stage meta-analysis of a GWAS, in addition to a replication cohort. Although the identification of SNPs associated with prognosis for breast cancer have the potential to have an impact on several clinical scenarios with respect to breast cancer treatment, the study is underpowered to find this association. No SNP formally reached genome-wide significant threshold, making the study preliminary. Accordingly, the results need to be further tempered. In its current format, it is a bit difficult to follow and it is not clear if the standard quality control metrics have been performed and reported. In other words, the method section is short on critical details. So, the functional inference is basically that and in need of stronger evidence to compensate for the lack of GW significance.

A series of key issues need to be either addressed or clarified.

1. It is not clear if the authors checked for population substructure. In the results section on page 5, the authors wrote that population stratification is unlikely to contribute to the significance of these SNPs, however, this is an important part of any GWAS analysis and thus population stratification must be checked and reported.
2. It is confusing to follow the sequence of validation studies. There are only three signals from the stage-2 analysis, since rs725212 and rs10963755 are not independent each other but yet the authors report in the results state 4 SNPs associated with OS/DPF. The paper must be corrected for 3, since two are the same signal.
3. The multivariable model almost reached genome-wide significance since 5.37×10^{-8} is higher than 5.0×10^{-8} . The authors must include the word 'almost' or something like that on this phrase. In the same direction, on the discussion, the authors should remove the statement that the four SNPs (but three signals and, thus, associations) 'approach genome wide levels of significance'.
4. The 'functional inference' section provides less data than the discussion section. For example, the AREG eQTL is not present in the results section, only on the discussion section. About that, how the authors can explain this eQTL result since the ADAMTSL1 expression does not correlate with rs715212?
5. The discussion section is full of potential mechanisms that does not makes sense without further data. Moreover, since most of the genes discussed does not showed eQTL results, the level of evidence is lacking and perhaps should focus on other approaches to elucidate an epigenetic mechanism- though there is no data to support this at present. The authors should expand their discussion on the limitation of the study due to sample size.
6. The criterias for selecting SNPs for stage-2, as written, are confusing. The authors said that 50 SNPs were selected based on their p-values and correlated SNPs. Then, additional 37 SNPs were selected based on their p-value, only. Why this differences? Looks like the authors are picking SNPs and losing the advantage of unbiased genome-wide analysis of the GWAS. At the end, the selected SNPs were located close to relevant genes, which reduced the chances of new discoveries. Clarification of these criteria and discussion on the limitation of these approach are lacking in the paper.

Reviewer #2 (Expert in cancer epidemiology):

This well-written paper examines over 6000 women with breast cancer to identify inherited variants affecting disease-free and overall survival. The authors identify 4 SNPs with suggestive association with prognosis, with one SNP as genome-wide significant. This is a novel finding, but with the caveat that the SNP is only genome-wide significant in a multivariable model, and only in a subset of 2293 early onset cases. However, given the paucity of genome-wide significant

findings for breast cancer prognosis, (despite some very large studies), this is a finding likely to be of interest to the field. The claim in the abstract, of association with AREG expression, is based on weak GTEx eQTL data presented in the discussion, so this assertion seems rather over-stated.

In general I would have liked to see more in the discussion to put this study into the context of previous reports, especially previous larger studies such as Guo et al 2015 (37954 cases) pmid 25890600 and Pirie 2015 (also over 37k cases) pmid 25897948. Is there any overlap of cases with these?

The rationale for a smaller less heterogeneous cohort is mentioned, but what are the limitations of the study? The authors give some power calculations and the study is powered for alpha 0.05, not genome-wide significance, so what are the false positive report probabilities of their findings?

There is a lot in the discussion about possible functional relevance but this is largely speculation without identification of the target gene of the SNPs, and could be cut down considerably.

Other comments:

Line 70: It is a bit contradictory to say "identified by gwas do not reach genome-wide significance"

Line 75: "suggest" rather than "provide" targets

Line 78: "larger component" than what?

Line 104: it would be helpful to have a sentence or two here describing the stage 1 and stage 2 cohorts here, or at least referring to the appropriate section of the methods

Line 111-line 115: this might be better in the methods, with a brief referral here?

Line 132: It would be good to see the lamda values quoted here

Line 138: the study only has sufficient power at alpha 0.05, so it is not surprising that no associations achieved genome-wide significance, so this should be re-phrased.

Line 139: "our selection criteria" needs explanation/reference here since this is the first we've heard of it

Line 140: I only saw 3.5×10^{-7} not 2.5×10^{-7}

Line 147-150: these seem rather arbitrary

Line 191: $p=0.015$ is not "very similar" to 1.3×10^{-4}

Line 243: I think it is misleading to use the term "approach genome-wide levels of significance" "suggestive levels" would better reflect reality

Line 417: Good to see the mean and SD for time between diagnosis and age at registration mentioned, but it would be better to correct for this left truncation in the survival analysis, as the SD is fairly high compared to some of the median survival times.

Fixed effects analysis is used throughout and this is probably Ok based on Q values in table 2, but justification for fixed effects rather than random effects should be mentioned.

Most of the eQTL results mentioned seem statistically rather weak, and are there other stronger eQTLs in the same region?

Table 1 why aren't exact p values given for all comparisons, some are <0.0001 , some are 1.49×10^{-11} for example. Please clarify if 5 groups have been compared, rather than stage 1 vs stage 2

Reviewers' comments:

Reviewer #1 (Expert in cancer genomics):

Kadalayil et al. investigated SNPs associated with breast cancer prognosis using a two-stage meta-analysis of a GWAS, in addition to a replication cohort. Although the identification of SNPs associated with prognosis for breast cancer have the potential to have an impact on several clinical scenarios with respect to breast cancer treatment, the study is underpowered to find this association. No SNP formally reached genome-wide significant threshold, making the study preliminary. Accordingly, the results need to be further tempered. In its current format, it is a bit difficult to follow and it is not clear if the standard quality control metrics have been performed and reported. In other words, the method section is short on critical details. So, the functional inference is basically that and in need of stronger evidence to compensate for the lack of GW significance.

The study involves a meta-analysis of four different GWAs with a combined sample size of 6042 patients with 1101 OS events and 1316 DFS events. It is also the largest study to date of early onset breast cancer (n=2315 patients aged ≤ 40 at diagnosis) which is a rare subgroup (7% of patients are aged 40 or less at diagnosis) that is particularly relevant to studies of breast cancer survival given the worse prognosis in this group. Meta-analysis of stage 1 and 2 for DFS has 80% power to detect common SNPs (MAF=0.3) with effect sizes of 1.3 in all patients and 1.42 in the early onset subset at $p=5 \times 10^{-8}$. Thus we consider our study to be well powered to detect associations such as rs715212 (Multivariable DFS HR=1.38, MAF=0.27). We have added this power calculation (stages 1 and 2 combined) to the discussion along with an acknowledgment that the study only has 20% power to detect rare SNPs (MAF=0.1) associated with DFS with HR=1.31 for all patients and HR=1.44 in early onset cases at $p=5 \times 10^{-8}$.

Although rs715212 did not quite reach genome-wide significance ($p=5.37 \times 10^{-8}$ instead of $\leq 5 \times 10^{-8}$) it is associated in 3 independent cohorts including the replication dataset, there is no evidence for heterogeneity between cohorts, it is assessed as noteworthy according to the false positive report probability (FPRP<0.2) and, to our knowledge, it represents the third most significant association with breast cancer survival. We consider this to be a robust finding that will be of interest to the field, as noted by the second reviewer, despite the tentative functional evidence.

All of the standard quality control measures are reported in the methods section (eg MAF, missingness per SNP and per individual, HWE, strand issues, gender, relatedness, population stratification and QQ plots).

A series of key issues need to be either addressed or clarified.

1. It is not clear if the authors checked for population substructure. In the results section on page 5, the authors wrote that population stratification is unlikely to contribute to the significance of these SNPs, however, this is an important part of any GWAS analysis and thus population stratification must be checked and reported.

Evidence for population stratification was assessed by multidimensional scaling (MDS) analysis and individuals with inferred non European ancestry were excluded. The remaining individuals used for GWAs formed a single cluster which overlapped with the Caucasian reference. This

process is reported in the methods (sub heading Genotyping and quality control at stage-1) which also refer to the MDS plot (supplementary figure 1). To draw attention to this and the other important quality control measures a reference to this section has been added to the results.

“Following standard quality control (see methods)...”

Genomic inflation factors have also been added to the QQ plots which are low ($\lambda \leq 1.05$) and therefore support our interpretation of them that systematic biases such as population stratification are unlikely to contribute towards SNP significance.

“For each cohort, the QQ plots and low genomic inflation factors ($\lambda \leq 1.05$)...”

2. It is confusing to follow the sequence of validation studies. There are only three signals from the stage-2 analysis, since rs725212 and rs10963755 are not independent each other but yet the authors report in the results state 4 SNPs associated with OS/DPF. The paper must be corrected for 3, since two are the same signal.

We thank the reviewer for pointing this out. The text has been revised to show that three independent signals have been identified.

Results

“After testing for association with OS and DFS, we identified three independent signals that were represented by four SNPs (1. rs715212 and rs10963755, 2. rs12302097, 3. rs410155)...”

Discussion

“Although no variants were identified at a genome-wide level of significance, three signals had suggestive levels of significance (1. rs715212 and rs10963755, 2. rs12302097 and 3. rs410155)...”

“Our meta-analysis identified three independent signals associated with breast cancer prognosis that are independent of the classical prognostic factors.”

3. The multivariable model almost reached genome-wide significance since 5.37×10^{-8} is higher than 5.0×10^{-8} . The authors must include the word ‘almost’ or something like that on this phrase. In the same direction, on the discussion, the authors should remove the statement that the four SNPs (but three signals and, thus, associations) ‘approach genome wide levels of significance’.

Corrections have been made to show that one SNP (rs715212) approached a genome-wide level of significance in the multivariable model. All other significance levels are now described as suggestive.

Abstract

“Most importantly, the association with rs715212 was noteworthy according to the false positive report probability ($FPRP < 0.2$) and close to a genome-wide level of significance ($P_{multivariable} = 5.37 \times 10^{-8}$) in the multivariable model.”

Results

“After adjustment for ER status, tumour grade, maximum tumour diameter, axillary nodal status and study, the association between rs715212 and risk of disease progression in patients with early onset approached a genome-wide level of significance ($P_{multivariate}=5.37 \times 10^{-8}$, HR=1.38, Table 3).”

Discussion

“Although no variants were identified at a genome-wide level of significance, three signals had suggestive levels of significance (1. rs715212 and rs10963755, 2. rs12302097 and 3. rs410155) without heterogeneity between cohorts which included one noteworthy association (rs715212 FPRP<0.2). The signal involving rs715212 and rs10963755 was associated exclusively with disease progression in patients with early onset and, in a pooled analysis with adjustment for tumour characteristics and study, rs715212 approached a genome-wide level of significance.”

4. The ‘functional inference’ section provides less data than the discussion section. For example, the AREG eQTL is not present in the results section, only on the discussion section. About that, how the authors can explain this eQTL result since the ADAMTSL1 expression does not correlate with rs715212?

The AREG eQTL analysis has been moved to the results section. AREG was tested as a possible eQTL because of its relationship with ADAM17, which is related to ADAMTSL1, and because previous studies showed that AREG is differentially expressed between pre and postmenopausal women which fits with the age specific association of rs715212. The mechanism of how rs715212 influences the expression of AREG is unknown as we now state in the abstract. However, it is possible for regulatory elements to be located on different chromosomes. Alternatively rs715212 may influence AREG expression by altering the functionality of ADAMTSL1 and not its expression.

5. The discussion section is full of potential mechanisms that does not makes sense without further data. Moreover, since most of the genes discussed does not showed eQTL results, the level of evidence is lacking and perhaps should focus on other approaches to elucidate an epigenetic mechanism- though there is no data to support this at present. The authors should expand their discussion on the limitation of the study due to sample size.

Most of the potential mechanisms have been removed although a small number of potentially relevant findings have been kept. A paragraph concerning the limitations of the study, including power to detect rare SNPs, has been added to the discussion.

“The current study has several limitations which must be noted. First, none of the survival analyses were adjusted for treatment and the ABCFS cohort could not be included in the analyses of DFS, early onset and multivariable models because these variables were unavailable. Second, although the study is well powered to detect common SNPs we estimate that it has only 20% power to detect rare SNPs (MAF=0.1) associated with DFS with hazard ratios of 1.31 for all patients and 1.44 for those with early onset. Third, the associations with gene expression involve healthy participants and have relatively modest significance levels. Further analysis with larger sample sizes and adjustment for additional clinicopathological factors including treatment (chemotherapy and hormone therapy) may provide more information that could further improve survival analysis. Further functional studies involving breast cancer patients and

including epigenetic mechanisms should be performed to provide more insights about the three association signals identified in the present study.”

6. The criterias for selecting SNPs for stage-2, as written, are confusing. The authors said that 50 SNPs were selected based on their p-values and corralted SNPs. Then, additional 37 SNPs were selected based on their p-value, only. Why this differences? Looks like the authors are picking SNPs and losing the advantage of unbiased genome-wide analysis of the GWAS. At the end, the selected SNPs were located close to relevant genes, which reduced the chances of new discoveries. Clarification of these criteria and discussion on the limitation of these approach are lacking in the paper.

Completely unbiased methods of SNP selection have no means of excluding false positives which are likely to be among the most significant signals. They will also neglect moderately significant SNPs in favor of the most significant SNPs despite potentially overwhelming support from correlated SNPs and proximity to biologically relevant genes. We therefore used a published method (Tapper et al 2015, Nat Commun) which employed an unbiased approach to generate a shortlist of index SNPs with support from correlated SNPs. Priority, but not exclusivity, was then given to index SNPs that were close to a relevant gene. Two shortlists of index SNPs were made which used either a stringent or moderate set of p-value thresholds. SNPs were selected from the stringent shortlist first (n=50) and then from the moderate shortlist (n=37). Since priority but not exclusivity was given to SNPs close to relevant genes, 20 SNPs were selected on a completely unbiased basis and 67 were selected from the unbiased shortlist because they were close to a relevant gene. It is also important to note that the relationship between these relevant genes and breast cancer prognosis is poorly understood and not widely accepted. We therefore believe that our SNP selection criteria have not reduced the chances of new discoveries.

The following changes have been made to the methods section to improve their clarity.

“Secondly, 37 SNPs were selected with $P_{meta} \leq 0.01$ and support from correlated SNPs ($r2 \geq 0.5$, within 500kb, and survival $P_{meta} \leq 0.05$). In both cases, priority but not exclusivity, was given to SNPs that were located within or flanked by genes with functional relevance to breast cancer survival according to annotation from GeneAlacart (<https://genealacart.genecards.org/>). As a result, 67 of the 87 SNPs selected were located within or flanked by functionally relevant genes (Supplementary Table 2).”

Reviewer #2 (Expert in cancer epidemilogy):

This well-written paper examines over 6000 women with breast cancer to identify inherited variants affecting disease-free and overall survival. The authors identify 4 SNPs with suggestive association with prognosis, with one SNP as genome-wide significant. This is a novel finding, but with the caveat that the SNP is only genome-wide significant in a multivariable model, and only in a subset of 2293 early onset cases. However, given the paucity of genome-wide significant findings for breast cancer prognosis, (despite some very large studies), this is a finding likely to be of interest to the field. The claim in the abstract, of association with AREG expression, is based on weak GTEx eQTL data presented in the discussion, so this assertion seems rather over-stated.

We thank the reviewer for appreciating the potential relevance of this study and have revised the abstract to show that the association with AREG expression is tentative and further functional studies are required. The increased significance of the multivariable model suggests that tumour characteristics are likely to be confounders so they should be included in the adjusted model to determine a more accurate effect size of the SNP. Previous studies have shown that adjustment for prognostic factors will increase power (Hernandez et al 2004).

“Expression quantitative trait data suggested that rs715212 might influence the expression of AREG ($P_{eQTL}=0.035$), which is overexpressed in oestrogen receptor positive tumours from premenopausal women, although the mechanism is unknown and further functional studies are required for confirmation.”

In general I would have liked to see more in the discussion to put this study into the context of previous reports, especially previous larger studies such as Guo et al 2015 (37954 cases) pmid 25890600 and Pirie 2015 (also over 37k cases) pmid 25897948. Is there any overlap of cases with these?

To put this study into context, we have summarised the findings from previous larger studies (Guo et al 2015, Pirie et al 2015) and a smaller study (Li et al 2014) and noted that to our knowledge only two loci have previously been associated with breast cancer survival at a genome-wide level of significance. Although our sample size is much smaller than previous studies it has a significantly higher event rate, which is one of the main determinants of power. We discuss the evidence for previous findings in our own study and note that there is a small overlap between our cases and those used by Guo et al 2015 and Pirie et al 2015.

Discussion

“To our knowledge, only two loci (rs4458204 Li et al 2014 and rs2059614 Guo et al 2015) have confidently been associated with breast cancer survival at a genome-wide level of significance despite the analysis of up to 37,954 patients and an event rate of 7.6% (n=2900 deaths).”

“Although the present study has a comparatively small sample size (n=6042 patients including 2315 aged 40 or younger at diagnosis), the event rate is significantly higher over all patients (OS=18.2%, DFS=22.5%) and the early onset subset (OS=26.1%, DFS=30.9%) which is one of the main determinants of power. Consequently, the combined analysis of stages 1 and 2 is estimated to have 80% power to detect common SNPs (MAF=0.3) with an effect size of HR=1.31 in all patients and HR=1.42 in the early onset subset at a genome-wide level of significance.”

“The genome-wide significant SNPs identified by previous studies were both associated with survival in ER-negative patients. We therefore repeated the stage 1 meta-analysis using ER-negative patients only (n=1637) but failed to replicate the findings for rs4458204 ($P_{meta}=0.771$) and rs2059614 ($P_{meta}=0.482$) despite a small overlap in the patients tested by these studies (n=196 from HEBCS for rs2059614 and n=315 from POSH for rs4458204). However, the current study lacks power to replicate these findings given the small number of ER-negative patients and low minor allele frequency for rs4458204 (MAF=0.12) and rs2059614 (MAF=0.03).

In a subsequent study which aimed to replicate SNPs with suggestive significance levels using the same patient cohorts, Pirie et al 2015 identified twelve variants with nominal significance ($P<0.05$) including seven that were associated with ER-positive disease. Eleven of these SNPs were genotyped and tested by the current study which replicated the associations with

rs1800566 in all patients ($P_{meta}=0.01$) and rs10477313 in ER-positive patients ($P_{meta}=0.013$). The nine remaining variants showed no evidence of association although five of these had $MAF<0.1$.”

The rationale for a smaller less heterogeneous cohort is mentioned, but what are the limitations of the study? The authors give some power calculations and the study is powered for alpha 0.05, not genome-wide significance, so what are the false positive report probabilities of their findings?

The power calculations have been revised to show that the study has 80% power to detect common SNPs ($MAF=0.3$) with modest effect sizes ($HR=1.31$ to 1.7) at a genome wide level of significance.

Results

“According to these sample sizes and event rates, we estimated that the combined stage-1 analysis of OS and DFS had 80% power to detect common SNPs ($MAF=0.3$) with modest effects ($HR=1.4$, $P=5\times 10^{-8}$). Due to its smaller sample size, the analysis of OS and DFS in early onset cases was estimated to have 80% power to detect SNPs with slightly larger effect sizes (OS: $HR=1.7$, DFS: $HR=1.6$, Supplementary Figure 2).”

Discussion

“Consequently, the combined analysis of stages 1 and 2 for DFS is estimated to have 80% power to detect common SNPs ($MAF=0.3$) with an effect size of $HR=1.31$ in all patients and $HR=1.42$ in the early onset subset at a genome-wide level of significance.”

Methods

“The power to detect SNPs associated with OS and DFS in all cases and patients with early onset breast cancer in the combined stage-1 cohorts was estimated using the survSNP program in R (65) with an additive genetic risk model and type 1 error rate of 5×10^{-8} (Supplementary Figure 2).”

A paragraph has been added to the discussion regarding the limitations of the study which include low power to detect rare SNPs, exclusion of the ABCFS cohort from all survival analyses except OS in all patients, no adjustment for treatment and nominal significance levels for the associations with gene expression.

“The current study has several limitations which must be noted. First, none of the survival analyses were adjusted for treatment and the ABCFS cohort could not be included in the analyses of DFS, early onset and multivariable models because these variables were unavailable. Second, although the study is well powered to detect common SNPs we estimate that it has only 20% power to detect rare SNPs ($MAF=0.1$) associated with DFS with hazard ratios of 1.31 for all patients and 1.44 for those with early onset (Supplementary Figure 2). Third, the associations with gene expression involve healthy participants and have relatively modest significance levels. Further analysis with larger sample sizes and adjustment for additional clinicopathological factors including treatment (chemotherapy and hormone therapy) may provide more information that could further improve survival analysis. Further functional investigation including epigenetic mechanisms should be performed to provide more insights about the three association signals identified in the present study.”

False positive report probabilities for the meta-analysis of stages 1 and 2 have been added (Table 2). According to these values the association with rs715212 is noteworthy.

Abstract

“Most importantly, the association with rs715212 was noteworthy according to the false positive report probability (FPRP<0.2) and approached a genome-wide level of significance ($P_{multivariable}=5.37 \times 10^{-8}$) in the multivariable model.”

Results

“Furthermore, the association with rs715212 is considered to be noteworthy according to the false positive report probability (FPRP)≤0.2.”

Methods

“False positive report probability

To assess the reliability of the associations from meta-analysis of stages 1 and 2 we calculated the false-positive report probability (FPRP) which describes the probability of no true association between a genetic variant and disease, given a statistically significant finding (Wacholder et al. 2004). The FPRP was calculated using a low prior probability of 0.0001, which is expected for a random SNP, to detect a hazard ratio of 1.3. A threshold of FPRP ≤0.2 was used to identify noteworthy associations.”

Discussion

“Although no variants were identified at a genome-wide level of significance, three signals had suggestive levels of significance (1. rs715212 and rs10963755, 2. rs12302097 and 3. rs410155) without heterogeneity between cohorts which included one noteworthy association (rs715212 FPRP<0.2).”

There is a lot in the discussion about possible functional relevance but this is largely speculation without identification of the target gene of the SNPs, and could be cut down considerably.

Most of the speculation regarding functional relevance has been removed from the discussion although a small number of potentially relevant findings have been retained.

Other comments:

Line 70: It is a bit contradictory to say “identified by gwas do not reach genome-wide significance”

This sentence has been changed, the loci are now referred to as reported rather than identified.

“Most of these loci have small effect sizes (Hazard ratio (HR)<1.5) and it is important to note that many of the loci reported by GWAs did not reach genome-wide significance which suggests that some of the previous GWAs were under-powered due to small sample size.”

Line 75: “suggest” rather than “provide” targets

We have changed provide to suggest.

“These genetic determinants of prognosis are important because they could improve prognostic models, aid selection of appropriate treatments, and suggest targets for new therapies.”

Line 78: “larger component” than what?

We meant that the performance of tumour gene expression profiles might be because they account for a larger component of germline determinants of gene expression than models based on clinicopathologic features. We have revised this sentence as follows.

“For example, tumour gene expression profiles perform equally well or better than clinicopathologic models, possibly because they reflect a larger component of germline determinants of gene expression in an established tumour than models based on clinicopathologic features (10).”

Line 104: it would be helpful to have a sentence or two here describing the stage 1 and stage 2 cohorts here, or at least referring to the appropriate section of the methods

A sentence has been added to describe the stage 1 and 2 cohorts which refers to the methods section for more details.

“Stage-1 breast cancer samples came from four cohorts from Australia (ABCFS), Helsinki (HEBCS), the United Kingdom (POSH) and Germany (SUCCESS-A). A further 1303 independent patients from the POSH cohort were used for replication analysis at stage-2 (Table 1, see Methods for a full description of these cohorts).”

Line 111-line 115: this might be better in the methods, with a brief referral here?

We now give a brief summary regarding methods used to correct for differences between cohorts in the results section and have moved the details to methods.

Results

“To address these differences, survival analyses at stage 1 and 2 were adjusted for ER status and, for replicating SNPs, multivariable models were constructed using pooled data from stages 1 and 2.”

Methods

“To identify SNPs influencing prognosis we used the formetascore command in GenABEL (66) to perform Cox regression analyses of overall survival (OS) and disease-free survival (DFS), with correction for ER status which is the only variable that is recorded in all four cohorts, has the most complete data and is associated with survival (20).”

Line 132: It would be good to see the lamda values quoted here

Lambda values are now quoted in this sentence and they have been added to the QQ plots (Supplementary figures 3-5).

“For each cohort, the QQ plots and low genomic inflation factors ($\lambda \leq 1.05$) for OS and DFS in all cases and in the early onset subset...”

Line 138: the study only has sufficient power at alpha 0.05, so it is not surprising that no associations achieved genome-wide significance, so this should be re-phrased.

The power calculations have been revised using $\alpha=5 \times 10^{-8}$ which show that the study has 80% power to detect common SNPs (MAF=0.3) with modest effect sizes (HR=1.4 for all patients and HR=1.6 for early onset).

Results

“Despite meta-analysis of upto 4,739 patients and sufficient power to detect common SNPs (MAF=0.3) with modest effect sizes (HR=1.4 for all patients and HR=1.6 for early onset), no associations achieved genome-wide significance at stage-1.”

Line 139: “our selection criteria” needs explanation/reference here since this is the first we’ve heard of it

We have added a reference to the methods section which describe the SNP selection criteria.

“We therefore used our selection criteria (see Methods) to select 87 SNPs for assessment at stage-2...”

Line 140: I only saw 3.5×10^{-7} not 2.5×10^{-7}

This line has been corrected to show that the stage 1 p-values actually ranged from 3.5×10^{-7} to 0.008.

“We therefore used our selection criteria (see Methods) to select 87 SNPs for assessment at stage-2, with P-values from the stage-1 meta-analysis ranging from 3.5×10^{-7} to 0.008 (Supplementary table 2).”

Line 147-150: these seem rather arbitrary

The criteria used to select published SNPs for replication are somewhat arbitrary but we could not afford to test all SNPs and believe it is reasonable to exclude those with modest significance or in genes which we have previously tested in the POSH cohort. Other GWAs have used similar criteria when selecting SNPs for follow up.

Line 191: $p=0.015$ is not “very similar” to 1.3×10^{-4}

The pooled analysis result for rs410155 is now described separately as less significant and with a smaller effect.

“Excluding rs410155, the significance levels and effect sizes from the pooled analyses were very similar to those obtained from meta-analysis (rs715212 $P_{\text{pooled}}=1.94 \times 10^{-5}$, HR=1.28; rs10963755 $P_{\text{pooled}}=6.48 \times 10^{-4}$, HR=1.21; rs12302097 $P_{\text{pooled}}=1.65 \times 10^{-4}$, HR=1.28, Table 3). For rs410155, the pooled analysis was less significant and the hazard ratio was slightly smaller ($P_{\text{pooled}}=0.015$, HR=1.20).”

Line 243: I think it is misleading to use the term “approach genome-wide levels of significance” “suggestive levels” would better reflect reality

The significance levels from meta-analysis are now described as suggestive and the multivariable result for rs715212 only is described as approaching genome wide significance.

“Although no variants were identified at a genome-wide level of significance, three signals had suggestive levels of significance (1. rs715212 and rs10963755, 2. rs12302097 and 3. rs410155) without heterogeneity between cohorts which included one noteworthy association (rs715212 FPRP<0.2).

The signal involving rs715212 and rs10963755 was associated exclusively with disease progression in patients with early onset and, in a pooled analysis with adjustment for tumour characteristics and study, rs715212 approached a genome-wide level of significance.”

Line 417: Good to see the mean and SD for time between diagnosis and age at registration mentioned, but it would be better to correct for this left truncation in the survival analysis, as the SD is fairly high compared to some of the median survival times.

We have not applied left truncation because the survival analysis is already left truncated in the sense that survival times begin from the date of diagnosis rather than registration. Additionally, diagnosis and registration dates are the same for many patients.

Fixed effects analysis is used throughout and this is probably Ok based on Q values in table 2, but justification for fixed effects rather than random effects should be mentioned.

A justification for the use of fixed effects meta analysis has been added to the methods section and we found no evidence of heterogeneity using the Cochran Q statistic.

“A fixed effects meta-analysis was used under the assumption that SNPs have one true effect size and that any differences between studies were most likely to be due to sampling variation.”

Most of the eQTL results mentioned seem statistically rather weak, and are there other stronger eQTLs in the same region?

For each associated SNP and SNPs in weak LD, we tested association with flanking genes and genes with functional relevance and reported all nominal associations ($P < 0.05$). We have subsequently repeated this analysis and found no evidence of stronger eQTLs in the same region.

Table 1 why aren't exact p values given for all comparisons, some are < 0.0001 , some are 1.49×10^{-11} for example. Please clarify if 5 groups have been compared, rather than stage 1 vs stage 2

The stats package gave exact p-values for the Pearson's chi-squared test but did not report p-values less than 0.0001 for the nonparametric test used to compare continuous traits (age at diagnosis and survival times for OS and DFS). These tests were repeated in R but the p-values were less than 2.2×10^{-16} which is the minimum. We therefore present these p-values as $< 2.2 \times 10^{-16}$

¹⁶ in table 2. This threshold is used in other publications and p-values below this level are not very informative.

The comparisons are made between all cohorts with data which ranges from 3 for progesterone receptor to 5 for oestrogen receptor. This clarification has been added to the methods and as part of the legend in table 1.

Methods

“To assess the similarity of clinicopathologic features between all cohorts with data (n=3 to 5), Pearson’s chi-squared test was used for categorical traits and Kruskal-Wallis rank sum tests were used for continuous traits (Table 1).”

Table 1 legend

“P-value for comparison between all cohorts with data (n=3 to 5)...”

Reviewers' comments:

Reviewer #1 (Remarks to the Author):

The authors revised the paper to address some of the reviewer's questions but alas not all. The paper is improved, but there are some questions to be solved. First, the authors must clarify for the reader the limitation of an association that is almost significant only on multivariable analysis when it is not present by the regular association analysis. This is complex and really critical- especially since the authors presented the study as sufficiently well powered to detect common SNPs (rs715212 MAF = 0.273, CEU). The second major issue not adequately resolved is the rs715212 relationship with AREG is, which remains overstated. Further functional data is needed and they have not provided this. It is an association study at heart and the authors have to be far more measured in the presentation.

Summary of the questions answered by the authors:

1. The number of regions: the authors corrected the number of regions (from 4 to 3).
2. With respect to the significance and limitation of the study: the authors made clear that "no variants were identified at a genome-wide level of significance", although "the study is well powered to detect common SNPs". On the limitation of the study, the authors should include two phrases to make clear the difference of reaching genome-wide significance on the association analysis vs multivariate analysis.
3. The criteria for selecting SNPs for stage-2: the explanation for the question 6 was better than the section added to the text. This explanation should be added to the Methods section and include the published method (Tapper et al 2015, Nat Commun).
4. The 'functional inference' section was reduced, but it is still overstated. I understand the reason why the authors want to connect rs715212 to AREG. Even if it is methylation mechanism, it would first change ADAMTSL1 mRNA levels (or protein levels), and then change AREG levels. The hypothesis is for a trans-eQTL relationship between rs715212 to AREG. Trans-eQTL requires higher number of samples and are rare ('Systematic identification of trans eQTLs as putative drivers of known disease associations'; Nature Genetics 45, 1238-1243 (2013) doi:10.1038/ng.2756). Did the authors correct the 0.035 p-value for multiple comparisons? Do the authors have evidence for chromosome interaction between rs715212 and AREG directly? Without ADAMTSL1 and any functional assay, rs715212 eQTL with AREG seems rather an overstated and it is difficult to keep this relationship on the paper.

Reviewer #2 (Remarks to the Author):

Dr. Tapper and colleagues have addressed the issues raised.

Reviewers' comments:

Reviewer #1 (Remarks to the Author):

The authors revised the paper to address some of the reviewer's questions but alas not all. The paper is improved, but there are some questions to be solved. First, the authors must clarify for the reader the limitation of an association that is almost significant only on multivariable analysis when it is not present by the regular association analysis. This is complex and really critical- especially since the authors presented the study as sufficiently well powered to detect common SNPs (rs715212 MAF = 0.273, CEU).

Many previous studies have shown that adjusting for known prognostic factors will increase statistical power in multivariable analysis of time to event data (eg Kahan et al 2014). We have added text to the discussion to highlight the increase in significance and effect size between uni and multivariable analysis (eg rs715212: univariable $P=1.94 \times 10^{-5}$, HR=1.28 multivariable $P=5.37 \times 10^{-8}$, HR=1.38 Table 3). We also suggest that this difference is likely to be due to the increased power of multivariable analysis which adjusts for the confounding effects of tumour characteristics and enables more accurate estimation of effect sizes. The lack of GWAs significant results in univariable analysis despite sufficient statistical power is attributed to confounding factors that influence survival times and are not considered when estimating power.

Discussion

"Despite sufficient statistical power no variants were identified at a genome-wide level of significance which is likely to be due to the large number of factors that influence survival times, including phenotypic heterogeneity, tumour biology and treatment. However, three signals had suggestive levels of significance (1. rs715212 and rs10963755, 2. rs12302097 and 3. rs410155) without heterogeneity between cohorts including one that was estimated to be noteworthy according to the false positive report probability (rs715212 FPRP<0.2). With the exception of rs12302097, the effect size (HR) and significance of these associations became stronger after adjusting for tumour characteristics in a multivariable model. This suggests that tumour characteristics are confounding factors and that accounting for them in a multivariable model will increase the accuracy of effect size estimates. Previous studies have shown that similar adjustments for known prognostic factors will increase statistical power in the analysis of time-to-event outcomes⁴⁰ "

The second major issue not adequately resolved is the rs715212 relationship with AREG is, which remains overstated. Further functional data is needed and they have not provided this. It is an association study at heart and the authors have to be far more measured in the presentation.

Currently we have no additional functional data to support the relationship between rs715212 and AREG. All references to this association have therefore been revised in order to acknowledge the limited evidence and need for additional confirmatory studies.

Abstract

"Expression quantitative trait data provide tentative evidence that rs715212 might influence the expression of *AREG* ($P_{eQTL}=0.035$), which is overexpressed in oestrogen receptor positive tumours from premenopausal women. Additional functional studies are needed to confirm this association and elucidate a possible mechanism."

Discussion

“We have shown that rs715212 is nominally associated with the expression of *AREG* and that the chromatin surrounding rs715212 is predicted to be functional. These findings provide tentative evidence that rs715212 and/or linked variants may influence disease progression in early onset patients by altering the methylation or functionality of ADAMTSL1 and/or the expression of *AREG* either directly or via regulation of other members of the ADAMTS gene family. However, it is important to stress that further functional studies are required to verify the association between rs715212 and *AREG* expression and determine the biological mechanism.”

Summary of the questions answered by the authors:

1. The number of regions: the authors corrected the number of regions (from 4 to 3).
2. With respect to the significance and limitation of the study: the authors made clear that “no variants were identified at a genome-wide level of significance”, although “the study is well powered to detect common SNPs”. On the limitation of the study, the authors should include two phrases to make clear the difference of reaching genome-wide significance on the association analysis vs multivariate analysis.

A sentence has been added to the study limitations to reiterate that no variants achieved GWAs significance and that the most significant results were obtained by multivariable analysis. Previous studies have shown that adjusting for known prognostic factors increases statistical power. This explanation for the difference between the uni and multivariable results has also been added to the discussion as described above in response to major issue 1.

Discussion

“First, no variants were identified at a genome-wide level of significance and the most significant results were derived from a multivariable analysis which adjusted for the confounding effect of tumour characteristics.”

3. The criteria for selecting SNPs for stage-2: the explanation for the question 6 was better than the section added to the text. This explanation should be added to the Methods section and include the published method (Tapper et al 2015, Nat Commun).

We have used the explanation for question 6 to describe the SNP selection and cited the PLINK publication as the original source which developed the clumping procedure.

Methods

“Completely unbiased methods of SNP selection have no means of excluding false positives which are likely to be among the most significant signals. They will also neglect moderately significant SNPs in favour of the most significant SNPs despite potentially overwhelming support from correlated SNPs and proximity to biologically relevant genes. To select the most promising SNPs, we therefore used a clumping procedure in PLINK⁴⁶ to generate a shortlist of index SNPs with support from correlated SNPs (SNPs $r^2 \geq 0.5$, within 500kb). Priority, but not exclusivity, was then given to index SNPs that were close to a relevant gene according to annotation from GeneAlacart (<https://genealacart.genecards.org/>). Two shortlists of index SNPs were made which used either a stringent (index SNP $P_{meta} \leq 0.001$ and correlated SNP $P_{meta} \leq 0.01$) or moderate set of p-value thresholds (index SNP $P_{meta} \leq 0.01$ and correlated SNP $P_{meta} \leq 0.1$). SNPs were selected from the stringent shortlist first (n=50) and then from the moderate shortlist (n=37). Since priority but not exclusivity was given to SNPs close to relevant genes, 20 SNPs were selected on a completely unbiased basis and 67 were selected from the unbiased shortlist because they were close to a relevant gene (Supplementary Table 2).

4. The ‘functional inference’ section was reduced, but it is still overstated. I understand the reason

why the authors want to connect rs715212 to AREG. Even if it is methylation mechanism, it would first change ADAMTSL1 mRNA levels (or protein levels), and then change AREG levels. The hypothesis is for a trans-eQTL relationship between rs715212 to AREG. Trans-eQTL requires higher number of samples and are rare ('Systematic identification of trans eQTLs as putative drivers of known disease associations'; Nature Genetics 45, 1238–1243 (2013) doi:10.1038/ng.2756). Did the authors correct the 0.035 p-value for multiple comparisons? Do the authors have evidence for chromosome interaction between rs715212 and AREG directly? Without ADAMTSL1 and any functional assay, rs715212 eQTL with AREG seems rather an overstated and it is difficult to keep this relationship on the paper.

We have revised the manuscript to ensure that all references to the association between rs715212 and AREG expression are not overstated and a statement has been added to the discussion and abstract which calls for additional studies in order to verify the association with AREG expression (see response to major issue 2).

The functional inference section presents two results for rs715212, one from a published analysis of chromatin states and the other from the Genotype-Tissue Expression (GTEx) project. In our opinion these results are accurately described and have not been overstated.

Several hypotheses are suggested in the discussion (methylation of ADAMTSL1, altered functionality of ADAMTSL1 including alternative splicing, and aberrant expression of AREG either as a trans-eQTL, by altered function of ADAMTSL1 or by other members of the ADAMTS family that could be regulated by the expression or function of ADAMTSL1). We agree with the reviewer that ADAMTSL1 expression is expected to change if methylation is involved. Although there was no evidence for an association with expression this result is inconclusive because it involves healthy participants as described in the study limitations. It is also possible that rs715212 affects ADAMTSL1 by altering its functionality and not its expression. Therefore a trans-eQTL is not the only possible hypothesis.

The eQTL analysis tested rs715212 for association with ADAMTSL1 and AREG only. Therefore a correction for multiple tests is not appropriate and was not applied.

Results

"The chromatin surrounding rs715212 is characterised as an enhancer in breast myoepithelial primary cells (strongly enriched for TF binding sites, moderately enriched for DNase peaks and conserved elements), while rs10963755 maps to a quiescent region."

"We therefore used GTEx to perform further eQTL analysis, and found that rs715212 is nominally associated with the expression of AREG (PFastQTL=0.035) in breast mammary tissue (Supplementary Figure 6)."

Discussion

"Previous studies have shown that ADAMTSL1 is a component of the extracellular matrix (ECM) that may function in cell-cell or cell-matrix interactions or may regulate other *ADAMTS* proteases³³. Although *ADAMTSL1* is primarily expressed in skeletal muscle, it has been seen in other tissues including breast and methylation studies have shown that it is hypermethylated in ER positive breast cancer tumours^{34,35}.

We have shown that rs715212 is nominally associated with the expression of *AREG* and that the chromatin surrounding rs715212 is predicted to be functional. These findings provide tentative evidence that rs715212 and/or linked variants may influence disease progression in early onset patients by altering the methylation or functionality of ADAMTSL1 and/or the expression of *AREG* either directly or via regulation of other members of the ADAMTS gene family. However, it is important to stress that further functional studies are required to verify the association between rs715212 and *AREG* expression and determine the biological mechanism."

Reviewer #2 (Remarks to the Author):

Dr. Tapper and colleagues have addressed the issues raised.

REVIEWERS' COMMENTS:

Reviewer #1 (Remarks to the Author):

the authors have adequately addressed this reviewer's comments